# TOWARDS ROBUST GRAPH NEURAL NETWORKS AGAINST LABEL NOISE

## ABSTRACT

Massive labeled data have been used in training deep neural networks, thus label noise has become an important issue therein. Although learning with noisy labels has made great progress on image datasets in recent years, it has not yet been studied in connection with utilizing GNNs to classify graph nodes. In this paper, we propose a method, named LPM, to address the problem using Label Propagation (LP) and Meta learning. Different from previous methods designed for image datasets, our method is based on a special attribute (label smoothness) of graph-structured data, i.e., neighboring nodes in a graph tend to have the same label. A pseudo label is computed from the neighboring labels for each node in the training set using LP; meta learning is utilized to learn a proper aggregation of the original and pseudo label as the final label. Experimental results demonstrate that LPM outperforms state-of-the-art methods in graph node classification task with both synthetic and real-world label noise. Source code to reproduce all results will be released.

## 1 INTRODUCTION

Deep Neural Networks (DNNs) have achieved great success in various domains, but the necessity of collecting large amount of samples with high-quality labels is both expensive and time-consuming. To address this problem, cheaper alternatives have emerged. For example, the onerous labeling process can be completed on some crowdsourced system like Amazon Mechanical Turk [1]. Besides, we can collect labeled samples from web with search engines and social media. However, all these methods are prone to produce noisy labels of low quality. As is shown in recent research (Zhang et al., 2016b), an intractable problem is that DNNs can easily overfit to noisy labels, which dramatically degrades the generalization performance. Therefore, it is necessary and urgent to design some valid methods for solving this problem.

Graph Neural Networks (GNNs) have aroused keen research interest in recent years, which resulted in rapid progress in graph-structured data analysis (Kipf & Welling, 2016; Velickovic et al., 2017; Xu et al., 2018; Hou et al., 2019; Wang & Leskovec, 2020). Graph node classification is the most-common issue in GNNs. However, almost all the previous works about label noise focus on image classification problem and handling noisy labels in the task of graph node classification with GNNs has not been studied yet. Fortunately, most edges in the graph-structured datasets are intra-class edges (Wang & Leskovec, 2020), indicating that a node's label can be estimated by its neighbor nodes' labels. In this paper, we utilize this special attribute of graph data to alleviate the damages caused by noisy labels. Moreover, meta learning paradigm serves as a useful tool for us to learn a proper aggregation between origin labels and pseudo labels as the final labels.

The key contributions of this paper are as follows:

- To the best of our knowledge, we are the first to focus on the label noise existing in utilizing GNNs to classify graph nodes, which may serve as a beginning for future research towards robust GNNs against label noise.
- We utilize meta-learning to learn how to aggregate origin labels and pseudo labels properly to get more credible supervision instead of learning to re-weight different samples.

---

[1] https://www.mturk.com/

We experimentally show that our LPM outperforms state-of-the-art algorithms in utilizing GNNs to classify graph nodes with both synthetic and real-world label noise.

## 2 RELATED WORK

### 2.1 GRAPH NEURAL NETWORKS

To start, we use $\mathcal{G} = (\mathcal{V}, \mathcal{E}, \mathcal{X})$ to denote a graph whose nodes set is $\mathcal{V}$ and edges set is $\mathcal{E}$, and $\mathcal{X} \in R^{n \times d}$ is the input feature matrix, where $n$ denotes the number of nodes in the graph and $d$ is the dimension of the input feature vector of each node. We use $e_{u,v} \in \mathcal{E}$ to denote the edge that connects node $u$ and $v$. For each node $v \in \mathcal{V}$, its neighbor nodes set can be donated as $\mathcal{N}_v = \{u : e_{u,v} \in \mathcal{E}\}$. For node classification task, the goal of GNNs is to learn optimal mapping function $f(\cdot)$ to predict the class label $y_v$ for node $v$. Generally speaking, GNNs follows a framework including aggregation and combination in each layer. Different GNNs have proposed different ways of aggregation and combination. In general, the $k$-th layer of a GNN reads

$$a_v^{(k)} = Aggregate^{(k)}(\{h_u^{(k-1)} : u \in \mathcal{N}(v)\}), h_v^{(k)} = Combine^{(k)}(h_v^{(k-1)}, a_v^{(k)}), \quad (1)$$

where $h_v^{(k)}$ is the output for $k$-th layer of node $v$, $h_v^{(0)}$ is the input vector of node $v$.

### 2.2 LABEL PROPAGATION

In Label Propagation (LP), node labels are propagated and aggregated along the edges in the graph (Zhou et al., 2004; Zhu et al., 2005; Wang & Zhang, 2007; Karasuyama & Mamitsuka, 2013). There are some works which were designed to improve the performance of label propagation. For example, Gong et al. (2016) proposed a novel iterative label propagation algorithm which explicitly optimizes the propagation quality by manipulating the propagation sequence to move from simple to difficult examples; Zhang et al. (2020) introduces a triple matrix recovery mechanism to remove noise from the estimated soft labels during propagation. Label propagation has been applied in semi-supervised image classification task. For example, Gong et al. (2017) used a weighted K-nearest neighborhood graph to bridge the datapoints so that the label information can be propagated from the scarce labeled examples to unlabeled examples along the graph edges. Park et al. (2020) proposed a novel framwork to propagate the label information of the sampled data (reliable) to adjacent data along a similarity based graph. Compared to these methods, we utilize the intrinsic graph structure instead of handcrafted graph to propagate clean labels information, which is more reliable for graph-structured data. Besides, GNNs are utilized by us to extract features and classify nodes for graph-structured data.

### 2.3 META-LEARNING BASED METHODS AGAINST NOISY LABELS

Meta-learning aims to learn not only neural networks' weights, but also itself, such as hand-designed parameters, optimizer and so on (Andrychowicz et al., 2016; Finn et al., 2017). Several works have utilized meta-learning paradigm to deal with label noise. For example, Li et al. (2019) has proposed to find noise-tolerant model parameters by keeping the consistency between the output of teacher and student networks, and Li et al. (2017b) trains the teacher networks with samples with clean labels and then transfer the knowledge to student networks so that the student can learn correctly even if the existence of mislabeled data. Besides, Ren et al. (2018); Jenni & Favaro (2018); Shu et al. (2019) utilize meta-learning paradigm to re-weight samples, i.e., weight samples with clean labels more and weight mislabeled samples less. The weighting factors are optimized by gradient decent or generated by a network to minimizes the loss on a small amount of samples with correct labels. In contrast, meta-learning paradigm is utilized in this paper to learn how to aggregate origin labels and pseudo labels properly. We can get more credible supervision by combining the original label information with the label information provided by LP properly.

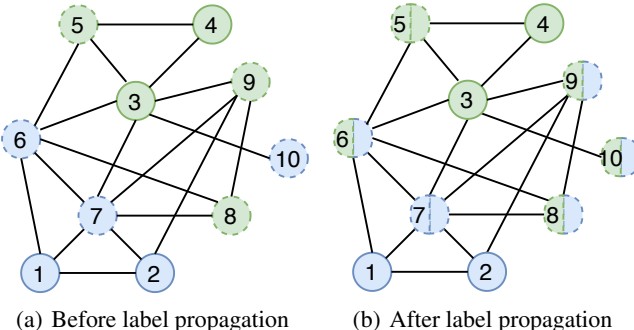

(a) Before label propagation          (b) After label propagation

Figure 1: Illustration of label propagation in our method. The two types of nodes are distinguished by two colours (blue and green). The nodes surrounded by dotted line are training nodes $\mathcal{D}_{train}$ whose label may be incorrect and those surrounded by solid line are clean sets $\mathcal{D}_{clean}$. In Figure.1(b), one half of every training node is pseudo label predicted by LP and the other half is original label. Some nodes' (node 5,7) pseudo labels are the same with their original labels, we select them $\mathcal{D}_{select}$ to train GNNs and inject them to clean sets for better label propagation. We can get proper labels for the left nodes $\mathcal{D}_{left}$ (node 6,8,9,10) based on meta learning.

## 3 METHODS

### 3.1 PRELIMINARIES

Given a graph data with $n$ nodes and their labels $\mathcal{D} = \{(x_0, y_0), (x_1, y_1), ..., (x_{n-1}, y_{n-1})\}$, where $x_j$ is the $j$-th node and $y_j \in \{0, 1\}^c$ is the label over $c$ classes. $\mathcal{D}_{train} = \{(x_0, y_0), (x_1, y_1), ..., (x_{s-1}, y_{s-1})\}$ are training nodes with noisy labels. Our goal is to enable the GNNs $f(x_j; w)$ trained with noisy sets $\mathcal{D}_{train}$ can also generalize well on test nodes. $w$ is the learnable parameters of GNNs. In our method, $m$ nodes with true labels $\mathcal{D}_{clean} = \{(x_s, y_s), (x_{s+1}, y_{s+1}), ..., (x_{s+m-1}, y_{s+m-1})\}$ in the graph are provided as the initial clean sets ($m \ll s$). GCN (Kipf & Welling, 2016) and GAT (Velickovic et al., 2017) are utilized in our experiments to extract features and classify nodes. Our method includes two main parts: label propagation and label aggregation. We will go into details about these two parts in the following section 3.2 and section 3.3.

### 3.2 LABEL PROPAGATION

Label Propagation is based on the label smoothness that two connected nodes tend to have the same label. Therefore, the weighted average of neighbor nodes' label of a node is similar to this node's true label. An illustration of LP part in our method can be found in Figure. 1. The first step of LP is to construct an appropriate neighborhood graph. A common choice is k-nearest graph (Iscen et al., 2019; Liu et al., 2018) but there is an intrinsic graph structure (adjacency matrix $A$) in graph data, so our similarities matrix $W$ with zero diagonal can be constructed with $A$, whose elements $W_{i,j}$ are pairwise similarities between node $i$ and node $j$:

$$W_{i,j} = \frac{A_{i,j}}{d(h_i, h_j) + \varepsilon}, \tag{2}$$

where $h_i, h_j$ are the feature vectors extracted by GNNs for node $i$ and node $j$. $d(\cdot, \cdot)$ is a distance measure (e.g.,Euclidean distance). $\varepsilon$ is an infinitesimal. Note that we can get $W$ with time complexity $\mathcal{O}(|\mathcal{E}|)$ instead of $\mathcal{O}(n^2)$ because $A$ is a sparse matrix whose edge lists are given. Then we can normalize the similarities matrix $W$:

$$S = D^{-1/2} W D^{-1/2}, \tag{3}$$

where $D$ is a diagonal matrix with $(i,i)$-value to be the sum of the $i$-th row of $W$. Let $Y^{(k)} = [y_1^{(k)}, ..., y_n^{(k)}]^T \in \mathbb{R}^{n \times c}$ be the soft label matrix in LP iteration $k$ and the $i$-th row $y_i^{(k)}$ is the predicted label distribution for node $i$. When $k = 0$, the initial label matrix $Y^{(0)} = [y_1^{(0)}, ..., y_n^{(0)}]^T$

consists of one-hot label vectors for $i = s, s+1, ..., s+m-1$(i.e., initial clean sets) or zero vectors otherwise. The LP (Zhu et al., 2005) in iteration $k$ can be formulated as:

$$Y^{(k+1)} = SY^{(k)}, \tag{4}$$

$$y_i^{(k+1)} = y_i^{(0)}, \forall i \in [s, s+m-1] \tag{5}$$

In Eq. (4), every node's label in the $(k+1)$-th iteration equals the weighted average of its neighbor nodes' labels in $k$-th iteration. In this way, the clean sets propagate labels to the noisy training nodes according to normalized edge weights. And then in Eq. (5), the labels of clean sets nodes are reset to their initial values. The reason is that we can take full advantage of the tiny minority of clean nodes and in case that the effect of clean sets fade away.

Co-teaching (Han et al., 2018) and Co-teaching plus (Yu et al., 2019) have been proposed to train DNNs robustly against label noise. There are two DNNs which select samples with small loss from noisy training sets to train each other. Our method is similar to theirs to some extent because LP is utilized by us to select true-labeled samples from $\mathcal{D}_{train}$ for training. However, instead of taking the nodes with small loss as true-labeled nodes, **we select the nodes $\mathcal{D}_{select}$ whose original labels are same with pseudo labels for training**. Original labels of $\mathcal{D}_{select}$ are credible and we also inject them to initial clean sets $\mathcal{D}_{clean}$ for better LP in next epoch. This is why our method can achieve better performance even if few true-labeled nodes are provided.

### 3.3 META-LEARNING BASED LABEL AGGREGATION

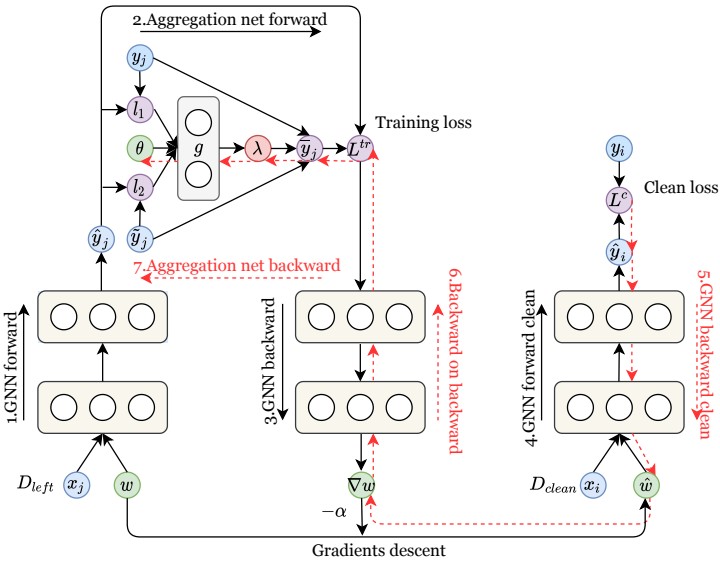

Figure 2: Computation graph of meta-learning based label aggregation.

In section 3.2, the selected training nodes (node 5,7 in Figure.1) have been utilized for training and LP but the left training nodes $\mathcal{D}_{left}$ (node 6,8,9,10 in Figure.1) with abundant information haven't been fully exploited. In this section, we mine the abundant and precious information from $\mathcal{D}_{left}$ via meta learning. The computation process of label aggregation is shown in Figure. 2.

For $\forall (x_j, y_j) \in \mathcal{D}_{left}$, we can get two loss values:

$$l_1 = loss(\hat{y}_j, y_j), \tag{6}$$

$$l_2 = loss(\hat{y}_j, \tilde{y}_j), \tag{7}$$

where $\hat{y}_j$ is the label predicted by GNNs for training node $j$ and $\tilde{y}_j$ is the pseudo label predicted by LP for node $j$. We can also get final label $\overline{y}_j$ for node $j$ by aggregating original label $y_j$ and pseudo label $\tilde{y}_j$:

$$\overline{y}_j = \lambda_j y_j + (1-\lambda_j)\tilde{y}_j, \lambda_j \in [0, 1] \tag{8}$$

where $\lambda$ is the aggregation coefficient. Some previous methods designed a weighting function mapping training loss to sample weights for noisy label problems (Kumar et al., 2010; Ren et al., 2018; Shu et al., 2019). Instead, we utilize a 3-layer multi-layer perceptron (MLP) as the aggregation network $g(\cdot; \cdot)$ to map loss values to aggregation coefficient $\lambda_j$:

$$\lambda_j = g(l_1 \| l_2; \theta) = \lambda_j(\theta; w), \tag{9}$$

Where $l_1 \| l_2$ is a 2-dimensional vector which is the concatenation of $l_1$ and $l_2$ and $\theta$ is the weights of aggregation network $g$. The rationality lies on a consensus that samples' loss values are affiliated with the credibility of samples' original labels (Kumar et al., 2010; Shu et al., 2019; Yu et al., 2019). The MLP or aggregation networks' input layer are 2 neurons and its output layer is one neuron, which can be an approximator to almost any continuous functions. The activation function of the last layer is sigmoid function to ensure that output $\lambda_j \in [0, 1]$. We can get the training loss $L_j^{tr}$ for node $j$:

$$L_j^{tr}(w, \theta) = loss(\hat{y}_j(w), \overline{y}_j(\theta)), \tag{10}$$

Then we can backward on the GNNs:

$$\hat{w}_t(\theta_t) = w_t - \frac{\alpha}{|\mathcal{D}_{left}|} \sum_{(x_j, y_j) \in \mathcal{D}_{left}} \nabla_w L_j^{tr}(w, \theta_t)|_{w_t}, \tag{11}$$

where $\alpha$ is the learning rate of GNNs. Then we can get the loss $L^c$ on clean sets $\mathcal{D}_{clean}$:

$$L^c(\hat{w}_t(\theta_t)) = \frac{1}{|\mathcal{D}_{clean}|} \sum_{(x_i, y_i) \in \mathcal{D}_{clean}} loss(f(x_i; \hat{w}_t(\theta_t)), y_i), \tag{12}$$

Where $f(x_i; \hat{w}_t(\theta_t))$ is the output of GNNs. Then we can utilize $L^c$ to update the weights of aggregation network:

$$\theta_{t+1} = \theta_t - \beta \nabla_\theta L^c(\hat{w}(\theta))|_{\theta_t}, \tag{13}$$

where $\beta$ is the learning rate of aggregation network. Finally, GNNs' weights can be updated:

$$w_{t+1} = w_t - \frac{\alpha}{|\mathcal{D}_{left}|} \sum_{(x_j, y_j) \in \mathcal{D}_{left}} \nabla_w L_j^{tr}(w, \theta_{t+1})|_{w_t}. \tag{14}$$

To some extent, this part is similar to re-weight based methods (Ren et al., 2018; Shu et al., 2019). However, LPM has two significant advantages. Firstly, re-weight based methods can not remove the damages caused by incorrect labels because they assign every noisy training sample a positive weight while LPM potentially has the ability to take full advantage of noisy samples positively. Secondly, LPM can generate comparatively credible labels for other usages while re-weight or some other methods can not. Algorithm. 1 shows all the steps of our algorithm.

### 3.4 Convergence of LPM

Here we show theoretically that the loss functions will converge to critical points under some mild conditions. The detailed proof of the following theorems will be provided in Appendix C.

**Theorem 1** *Suppose the loss function loss is L-Lipschitz smooth, and $\lambda(\cdot)$ is differential with a $\delta$-bounded gradient, twice differential with its Hessian bounded by $\mathcal{B}$ with respect to $\theta$. Let the learning rate $\alpha_t = \min\{1, \frac{k}{T}\}$, for some $k > 0$, such that $\frac{k}{T} < 1$ and learning rate $\beta_t$ a monotone descent sequence, $\beta_t = \min\{\frac{1}{L}, \frac{c}{\sqrt{T}}\}$ for some $c > 0$, such that $L \le \frac{c}{\sqrt{T}}$ and $\sum_{t=1}^{\infty} \beta_t \le \infty, \sum_{t=1}^{\infty} \beta_t^2 \le \infty$. Then the clean loss of Aggregation Net can achieve $\|\nabla_\theta L^c(\hat{w}(\theta_t))\|_2^2 \le \epsilon$ in $\mathcal{O}(1/\epsilon^2)$ steps. More specifically,*

$$\min_{0 \le t \le T} \|\nabla_\theta L^c(\hat{w}(\theta_t))\|_2^2 \le \mathcal{O}(\frac{C}{\sqrt{T}}). \tag{15}$$

**Theorem 2** *Under the conditions of Theorem 1, with the gradient of loss bounded by $\rho$, then*

$$\lim_{t \to \infty} \|\nabla_{w_t} L^{tr}(w_t, \theta_{t+1})\|_2^2 = 0. \tag{16}$$

---

**Algorithm 1:** LPM. Line 2-12: label propagation; Line 13-22: label aggregation.

---

**Data:** $\mathcal{D}, \mathcal{D}_{train}, \mathcal{D}_{clean}$, max epochs $T$, LP iterations $K$ in every epoch, $A$, feature matrix $\mathcal{X}$, GNNs feature extractor $f$, Aggregation Network $g$, expanding clean set for LP $\mathcal{D}_c$

**Result:** Robust GNNs parameters $w_T$

1   $\mathcal{D}_c = \mathcal{D}_{clean}$
2   **for** $t = 0, 1, 2, ..., T-1$ **do**
3     **for** $\forall v \in \mathcal{D}$ **do** $h_v = f(x_v; w_t)$;
4     **for** $(i, j) \in \{1, 2, ..., n\}^2$ **do** $W_{i,j} = \frac{A_{i,j}}{d(h_i, h_j) + \varepsilon}$ ;
5     **for** $k = 0, 1, 2, ..., K-1$ **do**
6       $Y^{(k+1)} = D^{-1/2} W D^{-1/2} Y^{(k)}, y_j^{(k+1)} = y_j^{(0)} (\forall \text{ node } j \in \mathcal{D}_c)$
7     **end**
8     $\mathcal{D}_{select} = \mathcal{D}_{left} = \varnothing$;
9     **for** $\forall \text{ node } i \in \mathcal{D}_{train}$ **do**
10       **if** $onehot(y_i^{(K)}) = y_i$ **do** $\mathcal{D}_{select} = \text{node } \{i\} \cup \mathcal{D}_{select}$;
11       **else do** $\mathcal{D}_{left} = \text{node } \{i\} \cup \mathcal{D}_{left}$;
12     **end**
13     $\mathcal{D}_c = \mathcal{D}_c \cup \mathcal{D}_{select}$
14     $w_t \leftarrow$ one-step optimization of $w_t$ with the selected nodes $\mathcal{D}_{select}$ ;
15     **for** $\forall \text{ node } j \in \mathcal{D}_{left}$ **do**
16       $\hat{y}_j = f(x_j; w_t)$;
17       $l_1 = loss(\hat{y}_j, y_j); l_2 = loss(\hat{y}_j, \tilde{y}_j)$;
18       $\lambda_j = g(l_1 \| l_2; \theta_t)$;
19       $\overline{y}_j = \lambda_j y_j + (1 - \lambda_j)\tilde{y}_j, \lambda_j \in [0, 1]; L_j^{tr}(w, \theta) = loss(\hat{y}_j(w), \overline{y}_j(\theta))$;
20     **end**
21     $\hat{w}_t(\theta_t) = w_t - \frac{\alpha}{|\mathcal{D}_{left}|} \sum_{(x_j, y_j) \in \mathcal{D}_{left}} \nabla_w L_j^{tr}(w, \theta_t)|_{w_t}$;
22     $L^c(\hat{w}_t(\theta_t)) = \frac{1}{|\mathcal{D}_{clean}|} \sum_{(x_i, y_i) \in \mathcal{D}_{clean}} loss(f(x_i; \hat{w}_t(\theta_t)), y_i)$;
23     $\theta_{t+1} = \theta_t - \beta \nabla_\theta L^c(\hat{w}(\theta))|_{\theta_t}$;
24     $w_{t+1} = w_t - \frac{\alpha}{|\mathcal{D}_{left}|} \sum_{(x_j, y_j) \in \mathcal{D}_{left}} \nabla_w L_j^{tr}(w; \theta_{t+1})|_{w_t}$.
25 **end**

---

## 4   EXPERIMENTS

### 4.1   DATASETS AND IMPLEMENTATION DETAILS

We validate our method on six benchmark datasets, namely citation networks (Sen et al., 2008) including **Cora**, **Citeseer** and **Pubmed**. **Coauthor-Phy** dataset (Shchur et al., 2018) is also utilized in our experiments, but the results are shown in Appendix A due to the limited space. Summary of the graph datasets mentioned above are shown in Table. 1. The **Clothing1M** (Xiao et al., 2015) and **Webvision** (Li et al., 2017a) dataset are utilized to validate the effectiveness of our method in real-world label noise settings. We take a $k$NN graph ($k = 5$) as the graph structure so that GNNs can be applied in these two datasets, which follows previous work (Franceschi et al., 2019). More details about our preprocessing on Clothing1M and Webvision datasets can be seen in Appendix B.

The experiments are conducted with two types of label noise: *uniform noise* and *flip noise* following previous works (Zhang et al., 2016a; Shu et al., 2019). The former means that the label of each sample is independently changed to a random class with probability $p$, and the latter means that the label is independently flipped to a similar class with total probability $p$. The ratio of training, validation, and test nodes are set as 4:4:2. Only nearly 25 nodes with clean labels in the validation set are provided as the clean set in each dataset and we ensure that each class has the same number of samples. For example, we use 8 clean samples per label class for Pubmed. GCN (Kipf & Welling, 2016) serves as the base classification network model in our experiments and it is trained using Adam (Kingma & Ba, 2014) with an initial learning rate 0.01 and a weight decay $5 \times 10^{-4}$, except that the weight decay equals to 0 in Clothing1M and Coauthor-Phy datasets.

We compare LPM with multiple baselines using the same network architecture. These baselines are typical and some of them achieve state-of-the-arts performance on image datasets, which include:

Table 1: Dataset statistics after removing self-loops and duplicate edges (Wang & Leskovec, 2020)

|  | Cora | Citeseer | Pubmed | Coauthor-Phy |
|---|---|---|---|---|
| #nodes | 2708 | 3327 | 19717 | 34493 |
| #edges | 5278 | 4552 | 44324 | 247962 |
| #features | 1433 | 3703 | 500 | 8415 |
| #classes | 7 | 6 | 3 | 5 |
| #Intra-class edge rate | 81.0% | 73.6% | 80.2% | 93.1% |

Table 2: Comparison with baselines in test accuracy (%) on Cora and Citeseer with uniform noise ranging from 0% to 80%. Mean accuracy (std) over 5 repetitions are reported. The best and the second best results are highlighted in **bold** and ***italic bold*** respectively.

| Datasets | Cora | | | | | Citeseer | | | | |
|---|---|---|---|---|---|---|---|---|---|---|
| Method/noise rate | 0.0 | 0.2 | 0.4 | 0.6 | 0.8 | 0.0 | 0.2 | 0.4 | 0.6 | 0.8 |
| Basemodel | 87.84 (0.04) | 85.92 (0.10) | 82.42 (0.13) | 75.77 (0.18) | 56.32 (0.19) | 77.67 (0.13) | 76.06 (0.15) | 72.97 (0.09) | 67.98 (0.12) | 55.26 (0.22) |
| GCN+FT | 88.05 (0.06) | 86.07 (0.13) | 82.48 (0.14) | 75.88 (0.15) | 58.81 (0.22) | 77.86 (0.15) | 76.24 (0.07) | 73.42 (0.21) | 68.13 (0.19) | 56.12 (0.28) |
| L2RW | **88.84 (0.19)** | 85.10 (0.21) | 80.67 (0.22) | 73.43 (0.42) | 50.09 (0.37) | 76.73 (0.20) | 73.68 (0.14) | 69.93 (0.29) | 62.31 (0.32) | 46.55 (0.49) |
| Co-teaching plus + FT | 86.76 (0.14) | 83.03 (0.19) | 71.68 (0.21) | 50.05 (0.31) | 36.39 (0.44) | 76.28 (0.19) | 75.49 (0.24) | 72.71 (0.13) | 66.63 (0.41) | 56.27 (0.36) |
| MW-Nets | 88.33 (0.16) | 85.93 (0.22) | 82.61 (0.45) | 75.60 (0.41) | 56.37 (0.51) | **78.27 (0.12)** | ***76.62 (0.14)*** | ***74.25 (0.21)*** | 68.06 (0.25) | 56.53 (0.45) |
| GCEloss+FT | 87.87 (0.13) | 85.10 (0.09) | ***82.89 (0.07)*** | ***76.16 (0.15)*** | ***60.43 (0.21)*** | 78.01 (0.12) | 76.54 (0.09) | 74.06 (0.18) | ***69.18 (0.24)*** | ***58.48 (0.31)*** |
| APL+FT | 87.68 (0.08) | ***86.26 (0.05)*** | 82.01 (0.13) | 74.49 (0.19) | 58.72 (0.25) | 76.54 (0.08) | 74.32 (0.17) | 71.77 (0.15) | 66.78 (0.22) | 56.08 (0.34) |
| **Ours** | ***88.75 (0.07)*** | **87.46 (0.11)** | **83.95 (0.15)** | **79.66 (0.22)** | **63.38 (0.27)** | ***78.12 (0.13)*** | **77.07 (0.06)** | **75.19 (0.15)** | **70.05 (0.11)** | **61.71 (0.22)** |

**Base model**, referring to the GCN that directly trained on noisy training nodes; Meta-learning based methods **L2RW** (Ren et al., 2018), **MW-Nets** (Shu et al., 2019); Typical and effective method **Co-teaching plus** (Yu et al., 2019); Robust loss function against label noise **GCE loss** (Zhang & Sabuncu, 2018) and **APL** (Ma et al., 2020); The most recent method based on co-training **JoCoR** (Wei et al., 2020). For those baselines that don't need clean sets (**Base model**, **Co-teaching plus**, **GCE loss**, **JoCoR** and **APL**), we **finetune** (denoted by **FT** in this paper) them on the initial clean sets after the model was trained on training sets for a fair comparison. More experimental details about LPM and all baselines are available in the Appendix B.

## 4.2 RESULTS

Table. 2 shows the results on Cora and Citeseer with different levels of uniform noise ranging from 0% to 80%. Every experiment are repeated 5 times with different random seeds. Finally, we report the best test accuracy across all epochs averaged over 5 repetitions for each experiment. As can be seen in Table. 2, our method gets the best performance across all the datasets and all noise rates, except the second for 0% uniform noise rate. Our method performs even better when the labels are corrupted at high rate. Table. 3 shows the performance on Cora, Citeseer and Pubmed with different levels of flip noise ranging from 0% to 40%. It can be seen that our method also outperforms state-of-the-arts methods under flip noise across different noise rate, except that the second for 0% flip noise rate. Our method outperforms the corresponding second best method by a large margin when the noise rate is 0.4. As can be seen in Table. 4, our method can also perform better than other baselines in datasets with real-world label noise. We also experiment with Graph Attention Networks (Velickovic et al., 2017) as the feature extractor and classifier, the results shown in Appendix A demonstrate that our method can also perform well with other GNNs.

Table 3: Comparison with baselines in test accuracy (%) on Cora , Citeseer and Pubmed with flip noise ranging from 0% to 40%. Mean accuracy (std) over 5 repetitions are reported. The best and the second best results are highlighted in **bold** and ***italic bold*** respectively.

| Datasets | Cora | | | Citeseer | | | Pubmed | | |
|---|---|---|---|---|---|---|---|---|---|
| Method/noise rate | 0 | 0.2 | 0.4 | 0 | 0.2 | 0.4 | 0 | 0.2 | 0.4 |
| Basemodel | 87.84 (0.04) | 81.64 (0.11) | 61.12 (0.24) | 77.67 (0.13) | 75.91 (0.14) | 52.67 (0.35) | 86.18 (0.08) | 85.30 (0.21) | 74.21 (0.29) |
| GCN+FT | 88.05 (0.06) | 82.89 (0.14) | 67.39 (0.42) | 77.86 (0.15) | 75.08 (0.22) | 61.41 (0.23) | 86.21 (0.09) | ***85.55 (0.24)*** | ***80.88 (0.32)*** |
| L2RW | **88.84 (0.19)** | 80.90 (0.21) | 59.00 (0.34) | 76.73 (0.20) | 71.85 (0.25) | 50.04 (0.44) | ***86.34 (0.14)*** | 84.54 (0.19) | 76.97 (0.31) |
| Co-teaching plus+FT | 86.76 (0.14) | 81.37 (0.21) | 53.00 (0.51) | 76.28 (0.19) | 74.66 (0.21) | 60.59 (0.33) | 85.59 (0.09) | 84.61 (0.22) | 73.99 (0.33) |
| MW-Nets | 88.33 (0.16) | ***85.33 (0.23)*** | 67.71 (0.43) | **78.27 (0.12)** | **76.84 (0.19)** | 61.97 (0.33) | 86.02 (0.07) | 84.74 (0.17) | 78.59 (0.28) |
| GCEloss+FT | 87.87 (0.13) | 83.21 (0.13) | 67.80 (0.37) | 78.01 (0.12) | 76.36 (0.20) | ***63.66 (0.46)*** | 86.15 (0.11) | 85.47 (0.06) | 80.03 (0.42) |
| APL+FT | 87.68 (0.08) | 81.09 (0.14) | ***70.07 (0.19)*** | 76.54 (0.08) | 73.38 (0.13) | 60.81 (0.52) | 86.16 (0.05) | 85.52 (0.06) | 70.08 (0.16) |
| **Ours** | ***88.75 (0.07)*** | **86.95 (0.12)** | **78.97 (0.33)** | ***78.12 (0.13)*** | ***76.39 (0.14)*** | **69.71 (0.39)** | **86.48 (0.05)** | **85.58 (0.13)** | **83.15 (0.36)** |

Table 4: Comparison with baselines in test accuracy (%) on Clothing1M and Webvision. Mean accuracy (± std) over 5 repetitions are reported. The best is highlighted in **bold**.

| Methods | Basemodel | GCN+FT | L2RW | MW-Nets | GCEloss+FT | JoCoR+FT | **Ours** |
|---|---|---|---|---|---|---|---|
| Clothing1M | 35.83±0.03 | 38.05±0.13 | 53.5±0.08 | 54.15±0.23 | 56.9±0.08 | 56.3±0.12 | **57.35±0.11** |
| Webvision | 32.43±0.05 | 34.58±0.08 | 50.12±0.16 | 52.42±0.25 | 53.45±0.13 | 54.12±0.22 | **55.43±0.17** |

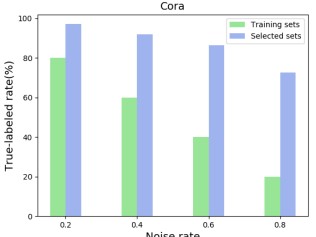 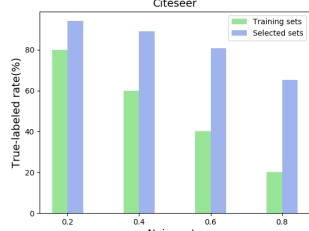 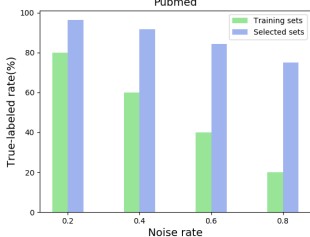

Figure 3: Comparsion of the true-labeled samples rate in $\mathcal{D}_{train}$ and $\mathcal{D}_{select}$ in various datasets.

Table 5: The performance of LPM without label aggregation and LPM with random $\lambda$ in Citeseer.

| Noise type | Uniform noise | | | | | Flip noise | |
|---|---|---|---|---|---|---|---|
| Method/noise rate | 0 | 0.2 | 0.4 | 0.6 | 0.8 | 0.2 | 0.4 |
| Ours w/o label aggregation | 72.07 | 68.36 | 65.69 | 62.16 | 54.39 | 69.26 | 63.14 |
| Ours with random $\lambda$ | 76.88 | 75.08 | 72.07 | 68.28 | 57.40 | 74.89 | 68.30 |
| Ours with tuned $\lambda$ | 77.22 | 76.31 | 73.55 | 69.17 | 58.54 | 75.11 | 68.72 |
| **Ours** | **78.12** | **77.07** | **75.19** | **70.05** | **61.71** | **76.39** | **69.71** |

### 4.3 ANALYSIS OF THE NECESSITY AND EFFECTIVENESS OF DIFFERENT PARTS

We design five experiments to validate the necessity and effectiveness of different components of our algorithm. Firstly, we compare the ratio of true-labeled nodes in $\mathcal{D}_{select}$ with $\mathcal{D}_{train}$ in the last epoch to validate the effectiveness of LP. Figure. 3 shows the ratio of true-labeled nodes in $\mathcal{D}_{select}$ in the last epoch and $\mathcal{D}_{train}$ under uniform noise on various datasets. It can be found that nearly all the nodes selected by LP are true-labeled even if most training nodes are mislabeled, which demonstrates the great ability of LP to select true-labeled nodes from noisy training nodes. Secondly, we remove the label aggregation in LPM to validate its necessity and the result shows that the performance of our method become much worse without label aggregation. It is necessary to mine the potential information from the left noisy training nodes after LP selection. Besides, we validate the effectiveness by replacing the learned aggregation coefficients $\lambda$ with random

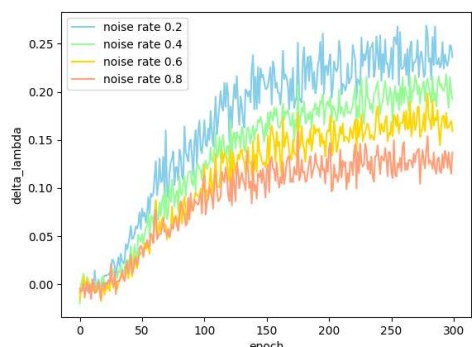

Figure 4: $\Delta\lambda$ varies during the training stage on Cora with various uniform noise rate.

numbers between 0 and 1. It is obvious that the aggregation coefficients $\lambda$ optimized by meta learning outperform random $\lambda$. Also, we assign the percentage of clean nodes of each label class as $\lambda$ (tuned) for comparison. These validate the effectiveness of the meta-learning based label aggregation. The results of above two experiments are shown in Table. 5. We denote the average of $\lambda$ of clean nodes and noisy nodes in $\mathcal{D}_{left}$ as $\lambda_{clean}$ and $\lambda_{noise}$ respectively, $\Delta\lambda = \lambda_{clean} - \lambda_{noise}$. We plot the variation of $\Delta\lambda$ during training stage in Figure. 4. It can be observed that $\lambda_{clean} > \lambda_{noise}$ across the training stage and the margin between $\lambda_{clean}$ and $\lambda_{noise}$ grows larger with the training process, which suggests that $\lambda$ optimized by our method is valid.

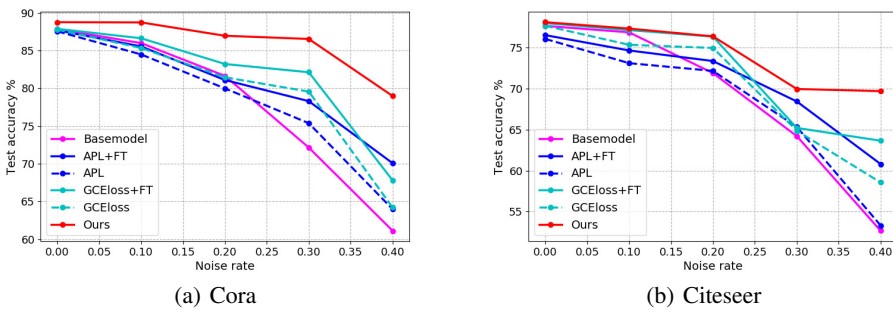

(a) Cora        (b) Citeseer

Figure 5: Test accuracy on Cora and Citeseer across various flip noise rate.

## 4.4 IMPACT OF FINETUNING AND NOISE RATE

We would like to investigate how our baselines can perform without finetuning. As can be seen in Figure. 5, the performance of the baselines will degenerate relatively significantly without finetuning across different noise rate. This illustrates that some baselines (without finetuning) that are designed for image datasets may perform relatively poor on graph-structured data and this motivates our work which trains GNNs robustly utilizing the structure information of graph data. Besides, We can also observe that our method only drops nearly 9% when the flip noise rate increased from 0% to 40%, whereas the baseline has dropped nearly 20% - 30%, which illustrates that our method is more robust, especially at high noise rate. At 0% noise, our method only slightly underperforms re-weights besed methods. This is reasonable because the original labels are all correct but our method will inevitably perturb a few clean labels while the re-weights based methods will not.

## 4.5 SIZE OF THE CLEAN SET

We try to strike a balance and understand when finetuning will be effective. As can be seen in Figure. 6, our method can also perform better even if the size of clean set is extremely small. The overall test accuracy does not grow much when the size of clean set is large enough. Besides, the test accuracy of baselines with fintuning will increase significantly when the size of clean set grows larger. This suggests that finetuning will be valid when the size of clean set grows larger because GNNs can achieve good performance with relatively less samples (Kipf & Welling, 2016; Veličković et al., 2017). From this perspective, our method can also serve as complementary for finetuning based methods when the size of clean set is large enough.

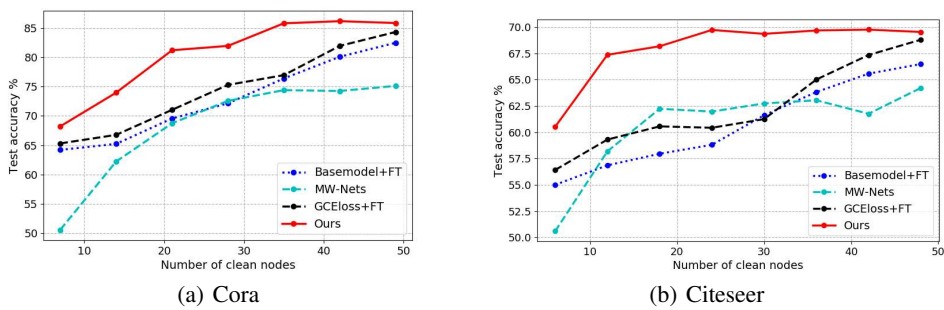

(a) Cora        (b) Citeseer

Figure 6: Test accuracy on Cora and Citeseer across various size of clean set.

## 5 CONCLUSION AND FUTURE WORK

In this work, we proposed a robust framwork for GNNs against label noise. This is the first method that specially designed for label noise problem existing in utilizing GNNs to classify graph nodes and it outperforms state-of-the-arts methods in graph-structured data, which may serve as the beginning for future research towards robust GNNs against label noise. As a future work, we may design an inductive robust method. Besides, better methods that don't need clean sets are also the goals of us.

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

## A    APPENDIX : ADDITIONAL EXPERIMENT RESULTS

Table A.6: Comparison with baselines in test accuracy (%) on Cora and Pubmed with flip noise ranging from 0% to 40% and **Graph Attention Networks**. The best result are highlighted in **bold**.

| Datasets | Cora | | | Pubmed | | |
|---|---|---|---|---|---|---|
| Methods/Noise rate | 0 | 0.2 | 0.4 | 0 | 0.2 | 0.4 |
| GAT | 89.85 | 84.13 | 67.10 | 85.55 | 84.57 | 74.11 |
| GAT+FT | 89.85 | 84.50 | 73.12 | 85.55 | 84.57 | 80.55 |
| MW-Nets | 87.52 | 84.26 | 69.99 | 85.64 | 84.5 | 75.82 |
| Co-teaching plus+FT | 88.56 | 85.42 | 74.94 | 85.56 | 84.48 | 82.40 |
| GCEloss+FT | 89.98 | 84.38 | 74.23 | 85.45 | 84.54 | 80.65 |
| JoCoR+FT | **90.16** | 85.00 | 73.74 | 85.47 | 84.58 | 80.95 |
| **Ours** | 89.92 | **87.20** | **75.65** | **85.72** | **84.62** | **83.00** |

Table A.7: Comparison with baselines in test accuracy (%) on Coauthor-Phy with flip noise ranging from 0% to 40%. The best result are highlighted in **bold**.

| Method/Noise rate | 0.0 | 0.1 | 0.2 | 0.3 | 0.4 |
|---|---|---|---|---|---|
| Basemodel | 96.92 | 96.32 | 95.57 | 94.91 | 86.25 |
| GCN+FT | 96.96 | 96.41 | 95.54 | 94.46 | 92.25 |
| Co-teaching plus+FT | 96.45 | 96.39 | 96.10 | 95.27 | 92.79 |
| MW-Nets | 96.56 | 96.24 | 95.62 | 95.56 | 89.25 |
| GCEloss+FT | **96.99** | 96.58 | 95.96 | 94.77 | 93.64 |
| JoCoR+FT | 96.83 | 96.59 | 96.07 | 94.95 | 94.11 |
| **Ours** | 96.75 | **96.71** | **96.49** | **96.14** | **95.14** |

We also take Graph Attention Networks (GAT) as the feature extractor and classifier and the results shown in Table. A.6 validate that our method can also perform well with various GNNs. Besides, LPM can also perform better than other baselines in larger graph dataset Coauthor-Phy, the results can be seen in Table. A.7. We also demonstrate confusion matrices of Basemodel and LPM in Figure. A.4, which visually show that our method can improve the robustness against label noise of GNNs by a large margin.

## B    APPENDIX : ADDITIONAL DETAILS OF OUR EXPERIMENTS

Original Clothing1M and Webvision datasets are all large-scale datasets with real-world label noise. We randomly choose 5000 images in 10 classes from original datasets and every image serves as a node in the graph, a kNN graph (k=5) is treated as the graph structure so that GNNs can be applied in Clothing1M datasets. This setting is similar to some previous works which also aim to apply GNNs in datasets without graph structure. ResNet-50 with ImageNet pretrained weights is utilized by us to extract feature vectors for all the images.

Table. A.8 shows the different hyper-parameters in LPM experiments for different datasets. In all the experiments, 25 true-labeled nodes are utilized as the initial clean sets or as the samples for

Table A.8: The hyper-parameters of LPM in different datasets.

| | Cora | Citeseer | Pubmed | Coauthor-Phy | Clothing1M |
|---|---|---|---|---|---|
| Aggregation Net's learning rate | $1 \times 10^{-4}$ | $1 \times 10^{-4}$ | $1 \times 10^{-3}$ | $1 \times 10^{-3}$ | $1 \times 10^{-3}$ |
| Aggregation Net's mid-dimension | 64 | 100 | 100 | 64 | 50 |
| Aggregation Net's weight decay | $1 \times 10^{-4}$ | $1 \times 10^{-4}$ | $1 \times 10^{-4}$ | $1 \times 10^{-4}$ | $1 \times 10^{-4}$ |
| LPA iterations | 50 | 50 | 50 | 50 | 50 |

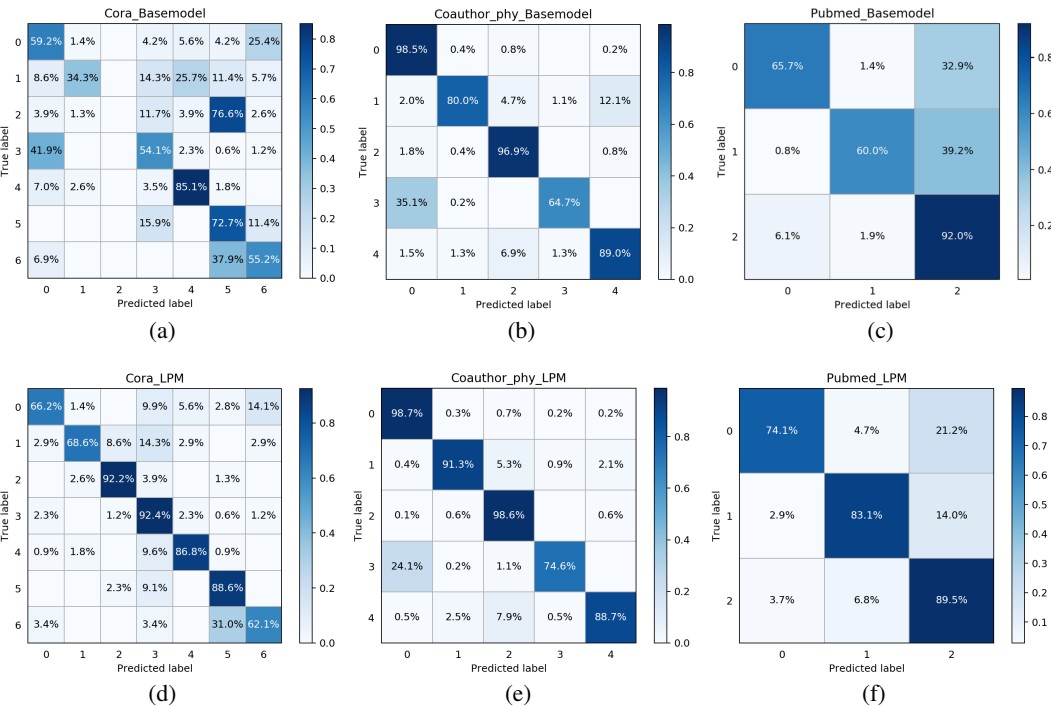

Figure A.4: Confusion matrices of Basemodel and LPM on various datasets under 40% flip noise. Figure. 4(a)-4(c) are the results of Basemodel. Figure. 4(d)-4(f) are the results of LPM.

finetuning and the total epoch of all the experiments is 300. In Co-teaching plus experiment, the initial epoch is 270, the forget rate is 0.1 and 5 epochs for linear drop rate ,the exponent of the forget rate is 1. For MW-Nets, the dimension of the meta net's middle layer is 100 and the learning rate is $5 \times 10^{-3}$. $q$ for GCEloss is 0.1. The combination of Normalized Focal Loss and Mean Absolute Error is utilized in APL experiments, the weight of Normalized Focal Loss is 0.1 and the weight of Mean Absolute Error is 10. For JoCoR experiments, the epochs for linear drop rate is 5 and the exponent of the forget rate is 2. The balance coefficient between conventional supervised learning loss and contrastive loss is 0.01. The learning rate and weight decay of Graph Attention Networks are 0.01 and $5 \times 10^{-4}$. The dimension of hidden layer of GAT is 16 and the number of head attentions is 8. The alpha of the leaky relu is 0.2 and the dropout rate is 0.5. Throughout this work we implemented gradient based meta-learning algorithms in PyTorch using the Higher library (Grefenstette et al., 2019).

## C   APPENDIX : CONVERGENCE OF LPM

Our proof of the convergence of LPM mainly follow some previous works (Ren et al., 2018; Shu et al., 2019) that utilize meta-learning to reweight noisy training samples. As is illustrated in some previous works (Zhou et al., 2004; Zhu et al., 2005), LPA will converge to a fixed point. Namely, $\mathcal{D}_{select}$ and $\mathcal{D}_{left}$ will converge to fixed sets. In our proof, the final $\mid \mathcal{D}_{left} \mid$ and final $\mid \mathcal{D}_{clean} \mid$ are denoted with $n$ and $m$ for easier illustration. Loss function $loss$ is denoted by $l$ in this proof. Here we first rewrite the forward and backward equations as follows:

$$\hat{y}_j = f(x_j; w_t) = y_j(w)|_{w_t} \tag{17}$$

$$\lambda_j = g(l(y_j, \hat{y}_j) \parallel l(\tilde{y}_j, \hat{y}_j); \theta_t) = \lambda_j(\theta; w_t)|_{\theta_t} \tag{18}$$

$$L^{tr}(w_t; \theta_t) = \frac{1}{n} \sum_{j=1}^{n} l(\lambda_j y_j + (1 - \lambda_j)\tilde{y}_j, \hat{y}_j) \tag{19}$$

$$\hat{w}_t(\theta_t) = w_t - \alpha \nabla_w L^c(w; \theta_t)|_{w_t} \tag{20}$$

$$\hat{y}_i = f(x_i; \hat{w}_t) = y_i(\hat{w}; x_i)|_{\hat{w}_t} \tag{21}$$

$$L^c(\hat{w})|_{\hat{w}_t} = \frac{1}{m}\sum_{i=1}^{m} L_i^c(\hat{w})|_{\hat{w}_t} = \frac{1}{m}\sum_{i=1}^{m} L_i^c(\hat{w}_t(\theta))|_{\theta_t} = \frac{1}{m}\sum_{i=1}^{m} l(y_i, \hat{y}_i) \tag{22}$$

$$\theta_{t+1} = \theta_t - \beta\nabla_\theta L^c(\hat{w}(\theta))|_{\theta_t} \tag{23}$$

$$w_{t+1} = w_t - \alpha\nabla_w L^{tr}(w; \theta_{t+1})|_{w_t} \tag{24}$$

$(x_j, y_j)$ is node from the final left training set $\mathcal{D}_{left}$;
$(x_i, y_i)$ is node from the final clean set $\mathcal{D}_{clean}$;
$f$ is the GCN for classification with its weights $w$;
$g$ is the Aggregation Net whose input are the nodes from clean set with its weights $\theta$;
$L^c$ is the loss on clean sets. $L^{tr}$ is the final training loss.

$l(y, \hat{y})$ is the loss (such as Cross Entropy) which satisfies linearity given by

$$l(\lambda y_1 + (1 - \lambda)y_2, \hat{y}) = \lambda l(y_1, \hat{y}) + (1 - \lambda)l(y_2, \hat{y}).$$

**Derivation of the equation of updating the weights in Aggregation Net**

$$\frac{1}{m}\sum_{i=1}^{m} \nabla_\theta L_i^c(\hat{w}(\theta))|_{\theta_t} = \frac{1}{m}\sum_{i=1}^{m} \frac{\partial L_i^c(\hat{w})}{\partial\hat{w}}|_{\hat{w}_t} \sum_{j=1}^{n} \frac{\partial\hat{w}_t(\theta)}{\partial\lambda_j}|_{\theta_t} \frac{\partial\lambda_j(\theta; w_t)}{\partial\theta}|_{\theta_t}. \tag{25}$$

According to Equation (20)

$$\hat{w}_t(\theta)|_{\theta_t} = w_t - \alpha\nabla_{w_t}\frac{1}{n}\sum_{j=1}^{n} l(\lambda_j y_j + (1 - \lambda_j)\tilde{y}_j, \hat{y}_j)$$

$$\frac{\partial\hat{w}_t(\theta)}{\partial\lambda_j}|_{\theta_t} = -\frac{\alpha}{n}\nabla_{w_t}\frac{\partial l(\lambda_j y_j + (1 - \lambda_j)\tilde{y}_j, \hat{y}_j)}{\partial\lambda_j}|_{\theta_t}$$

$$\frac{\partial\hat{w}_t(\theta)}{\partial\lambda_j}|_{\theta_t} = -\frac{\alpha}{n}\nabla_{w_t}\frac{\partial[\lambda_j l(y_j, \hat{y}_j) + (1 - \lambda_j)l(\tilde{y}_j, \hat{y}_j)]}{\partial\lambda_j}|_{\theta_t}$$

$$\frac{\partial\hat{w}_t(\theta)}{\partial\lambda_j}|_{\theta_t} = -\frac{\alpha}{n}\nabla_{w_t}(l(y_j, \hat{y}_j) - l(\tilde{y}_j, \hat{y}_j))|_{\theta_t}$$

$$\frac{\partial\hat{w}_t(\theta)}{\partial\lambda_j}|_{\theta_t} = -\frac{\alpha}{n}\frac{\partial(l(y_j, \hat{y}_j) - l(\tilde{y}_j, \hat{y}_j))}{\partial w_t}|_{w_t}$$

Therefore, Equation (25) can be written as

$$\frac{1}{m}\sum_{i=1}^{m} \nabla_\theta L_i^c(\hat{w}(\theta))|_{\theta_t}$$

$$= -\frac{\alpha}{mn}\sum_{i=1}^{m} \frac{\partial L_i^c(\hat{w})}{\partial\hat{w}}|_{\hat{w}_t} \sum_{j=1}^{n} \frac{\partial(l(y_j, \hat{y}_j) - l(\tilde{y}_j, \hat{y}_j))}{\partial w_t}|_{w_t} \frac{\partial\lambda_j(\theta; w_t)}{\partial\theta}|_{\theta_t}$$

$$= -\frac{\alpha}{n}\sum_{j=1}^{n} (\frac{1}{m}\sum_{i=1}^{m} \frac{\partial L_i^c(\hat{w})}{\partial\hat{w}}|_{\hat{w}_t}^T \frac{\partial(l(y_j, \hat{y}_j) - l(\tilde{y}_j, \hat{y}_j))}{\partial w_t}|_{w_t}) \frac{\partial\lambda_j(\theta; w_t)}{\partial\theta}|_{\theta_t}$$

$$= -\frac{\alpha}{n}\sum_{j=1}^{n} (\frac{1}{m}\sum_{i=1}^{m} G_{ij}) \frac{\partial\lambda_j(\theta; w_t)}{\partial\theta}|_{\theta_t},$$

where $G_{ij} = \frac{\partial L_i^c(\hat{w})}{\partial\hat{w}}|_{\hat{w}_t}^T \frac{\partial(l(y_j, \hat{y}_j) - l(\tilde{y}_j, \hat{y}_j))}{\partial w_t}|_{w_t}$.

**Lemma 1.** Suppose the loss function $l$ is L-Lipschitz smooth, and $\lambda(\cdot)$ is differential with a $\delta$-bounded gradient, twice differential with its Hessian bounded by $\mathcal{B}$ with respcet to $\theta$, and the loss function $l(\cdot, \cdot)$ have $\rho$-bounded gradients with respect to the parameter $w$. Then the gradient of $w$ with respect to $L_i^c(\hat{w})$ is Lipschitz continuous.

*Proof.* The supposition is equivalent to the following inequalities,

$$\|\nabla_{\hat{w}} L^c(\hat{w})|_{w_1} - \nabla_{\hat{w}} L^c(\hat{w})|_{w_2}\| \le L\|w_1 - w_2\|, \tag{26}$$

for any $w_1, w_2$;

$$\|\nabla_\theta \lambda(\theta; w_t)\| \le \rho; \tag{27}$$
$$\|\nabla_{\theta^2}^2 \lambda(\theta; w_t)\| \le \mathcal{B}; \tag{28}$$
$$\|\nabla_w l(y_i, \hat{y}_i((\hat{w}_t(w); x_i)))\| \le \delta. \tag{29}$$

The gradient of $\theta$ with respect to loss on clean set reads

$$\nabla_\theta L_i^c(\hat{w}(\theta))|_{\theta_t}$$
$$= -\frac{\alpha}{n} \sum_{j=1}^{n} \frac{\partial L_i^c(\hat{w})}{\partial \hat{w}}\Big|_{\hat{w}_t}^T \frac{\partial(l(y_j, \hat{y}_j) - l(\tilde{y}_j, \hat{y}_j))}{\partial w_t}\Big|_{w_t} \frac{\partial \lambda_j(\theta; w_t)}{\partial \theta}\Big|_{\theta_t}$$
$$= -\frac{\alpha}{n} \sum_{j=1}^{n} G_{ij} \frac{\partial \lambda_j(\theta; w_t)}{\partial \theta}\Big|_{\theta_t}$$

Taking the gradient of $\theta$ in both sides of the equation, we have

$$\nabla_{\theta^2}^2 L_i^c(\hat{w}(\theta))|_{\theta_t} = -\frac{\alpha}{n} \sum_{j=1}^{n} \left( \frac{\partial G_{ij}}{\partial \theta}\Big|_{\theta_t} \frac{\partial \lambda_j(\theta; w_t)}{\partial \theta}\Big|_{\theta_t} + (G_{ij}) \frac{\partial^2 \lambda_j(\theta; w_t)}{\partial \theta^2}\Big|_{\theta_t} \right).$$

For the first term in summation,

$$\Big\| \frac{\partial G_{ij}}{\partial \theta}\Big|_{\theta_t} \frac{\partial \lambda_j(\theta; w_t)}{\partial \theta}\Big|_{\theta_t} \Big\|$$
$$\le \delta \Big\| \frac{\partial}{\partial \hat{w}} \left( \frac{\partial L_i^c(\hat{w})}{\partial \theta}\Big|_{\theta_t}^T \right) \Big|_{\hat{w}_t}^T \frac{\partial(l(y_j, \hat{y}_j) - l(\tilde{y}_j, \hat{y}_j))}{\partial w_t}\Big|_{w_t} \Big\|$$
$$= \delta \Big\| \frac{\partial}{\partial \hat{w}} \left( -\frac{\alpha}{n} \sum_{k=1}^{n} \frac{\partial L_i^c(\hat{w})}{\partial \hat{w}}\Big|_{\hat{w}_t}^T \frac{\partial(l(y_k, \hat{y}_k) - l(\tilde{y}_k, \hat{y}_k))}{\partial w_t}\Big|_{w_t} \frac{\partial \lambda_j(\theta; w_t)}{\partial \theta}\Big|_{\theta_t} \right)\Big|_{\hat{w}_t}^T \frac{\partial(l(y_j, \hat{y}_j) - l(\tilde{y}_j, \hat{y}_j))}{\partial w_t}\Big|_{w_t} \Big\|$$
$$= \delta \Big\| \left( -\frac{\alpha}{n} \sum_{k=1}^{n} \frac{\partial^2 L_i^c(\hat{w})}{\partial \hat{w}^2}\Big|_{\hat{w}_t}^T \frac{\partial(l(y_k, \hat{y}_k) - l(\tilde{y}_k, \hat{y}_k))}{\partial w_t}\Big|_{w_t} \frac{\partial \lambda_k(\theta; w_t)}{\partial \theta}\Big|_{\theta_t} \right)\Big|_{\hat{w}_t}^T \frac{\partial(l(y_j, \hat{y}_j) - l(\tilde{y}_j, \hat{y}_j))}{\partial w_t}\Big|_{w_t} \Big\|$$
$$\le \delta \alpha \Big\| \frac{\partial^2 L_i^c(\hat{w})}{\partial \hat{w}^2}\Big|_{\hat{w}_t} \Big\| \Big\| \frac{\partial(l(y_k, \hat{y}_k) - l(\tilde{y}_k, \hat{y}_k))}{\partial w_t}\Big|_{w_t} \Big\| \Big\| \frac{\partial \lambda_k(\theta; w_t)}{\partial \theta}\Big|_{\theta_t}\Big|_{\hat{w}_t} \Big\| \Big\| \frac{\partial(l(y_j, \hat{y}_j) - l(\tilde{y}_j, \hat{y}_j))}{\partial w_t}\Big|_{w_t} \Big\|$$
$$\le 4\alpha L \rho^2 \delta^2.$$

And for the second term,

$$\Big\| (G_{ij}) \frac{\partial^2 \lambda_j(\theta; w_t)}{\partial \theta^2}\Big|_{\theta_t} \Big\| = \Big\| \frac{\partial L_i^c(\hat{w})}{\partial \hat{w}}\Big|_{\hat{w}_t}^T \frac{\partial(l(y_j, \hat{y}_j) - l(\tilde{y}_j, \hat{y}_j))}{\partial w_t}\Big|_{w_t} \frac{\partial^2 \lambda_j(\theta; w_t)}{\partial \theta^2}\Big|_{\theta_t} \Big\| \le 2\mathcal{B}\rho^2.$$

Therefore,

$$\|\nabla_{\theta^2}^2 L_i^c(\hat{w}(\theta))|_{\theta_t}\| \le 4\alpha^2 L \rho^2 \delta^2 + 2\alpha \rho^2 \mathcal{B}. \tag{30}$$

Let $L_v = 4\alpha^2 L \rho^2 \delta^2 + 2\alpha \rho^2 \mathcal{B}$, Based on Lagrange mean value theorem, we have

$$\|\nabla_\theta L^c(\hat{w}_t(\theta_1)) - \nabla_\theta L^c(\hat{w}_t(\theta_2))\| \le L_v \|\theta_1 - \theta_2\|,$$

for all $\theta_1, \theta_2$.

**Theorem 1.** Suppose the loss function $l$ is L-Lipschitz smooth, and $\lambda(\cdot)$ is differential with a $\delta$-bounded gradient, twice differential with its Hessian bounded by $\mathcal{B}$ with respect to $\theta$. Let the learning rate $\alpha_t = \min\{1, \frac{k}{T}\}$, for some $k > 0$, such that $\frac{k}{T} < 1$ and learning rate $\beta_t$ a monotone descent sequence, $\beta_t = \min\{\frac{1}{L}, \frac{c}{\sqrt{T}}\}$ for some $c > 0$, such that $L \leq \frac{c}{\sqrt{T}}$ and $\sum_{t=1}^{\infty} \beta_t \leq \infty, \sum_{t=1}^{\infty} \beta_t^2 \leq \infty$. Then the loss of Aggregation Net can achieve $\|\nabla_\theta L^c(\hat{w}(\theta_t))\|_2^2 \leq \epsilon$ in $\mathcal{O}(1/\epsilon^2)$ steps. More specifically,

$$\min_{0 \leq t \leq T} \|\nabla_\theta L^c(\hat{w}(\theta_t))\|_2^2 \leq \mathcal{O}(\frac{C}{\sqrt{T}}). \tag{31}$$

*Proof.* The iteration for updating the parameter $\theta$ reads

$$\theta_{t+1} = \theta_t - \beta \nabla_\theta L^c(\hat{w}_t(\theta))|_{\theta_t}.$$

In two successive iteration, observe that

$$L^c(\hat{w}_{t+1}(\theta_{t+1})) - L^c(\hat{w}_t(\theta_t))$$
$$= [L^c(\hat{w}_{t+1}(\theta_{t+1})) - L^c(\hat{w}_t(\theta_{t+1}))] + [L^c(\hat{w}_t(\theta_{t+1})) - L^c(\hat{w}_t(\theta_t))]. \tag{32}$$

For the first term, given that loss function on clean set is Lipschitz smooth, we have

$$L^c(\hat{w}_{t+1}(\theta_{t+1})) - L^c(\hat{w}_t(\theta_{t+1}))$$
$$\leq\, <\nabla L^c(\hat{w}_t(\theta_{t+1})), \hat{w}_{t+1}(\theta_{t+1}) - \hat{w}_t(\theta_{t+1})> + \frac{L}{2}\|\hat{w}_{t+1}(\theta_{t+1}) - \hat{w}_t(\theta_{t+1})\|_2^2.$$

According to Equation (20) and (23),

$$\hat{w}_{t+1}(\theta_{t+1}) - \hat{w}_t(\theta_{t+1}) = -\frac{\alpha_t}{n}\sum_{j=1}^{n}[\lambda_j \nabla_w l(y_j, \hat{y}_j) + (1-\lambda_j)\nabla_w l(\tilde{y}_j, \hat{y}_j)]|_{w_{t+1}},$$

and thus,

$$\|L^c(\hat{w}_{t+1}(\theta_{t+1})) - L^c(\hat{w}_t(\theta_{t+1}))\| \leq \alpha_t \rho^2 + \frac{L}{2}\alpha_t^2 \rho^2,$$

since the first gradient of loss function is bounded by $\rho$.

By the Lipschitz continuity of $L^c(\hat{w}_t(\theta))$ according to **Lemma 1.**, it can be obtained that

$$L^c(\hat{w}_t(\theta_{t+1})) - L^c(\hat{w}_t(\theta_t))$$
$$\leq \langle \nabla_{\theta_t} L^c(\hat{w}_t(\theta_t)), \theta_{t+1} - \theta_t \rangle + \frac{L}{2}\|\theta_{t+1} - \theta_t\|_2^2$$
$$= \langle \nabla_{\theta_t} L^c(\hat{w}_t(\theta_t)), -\beta_t \nabla_{\theta_t} L^c(\hat{w}_t(\theta_t)) \rangle + \frac{L\beta_t^2}{2}\|\nabla_{\theta_t} L^c(\hat{w}_t(\theta_t))\|_2^2$$
$$= -(\beta_t - \frac{L\beta_t^2}{2})\|\nabla_{\theta_t} L^c(\hat{w}_t(\theta_t))\|_2^2.$$

Therefore, the Equation (32) satisfies

$$L^c(\hat{w}_{t+1}(\theta_{t+1})) - L^c(\hat{w}_t(\theta_t)) \leq \alpha_t \rho^2 + \frac{L}{2}\alpha_t^2 \rho^2 - (\beta_t - \frac{L\beta_t^2}{2})\|\nabla_{\theta_t} L^c(\hat{w}_t(\theta_t))\|_2^2$$
$$(\beta_t - \frac{L\beta_t^2}{2})\|_2^2 \nabla_{\theta_t} L^c(\hat{w}_t(\theta_t))\|_2^2 \leq \alpha_t \rho^2 + \frac{L}{2}\alpha_t^2 \rho^2 - L^c(\hat{w}_{t+1}(\theta_{t+1})) + L^c(\hat{w}_t(\theta_t)).$$

Summing up above inequalities from $1$ to $T$, we have

$$\sum_{t=1}^{T}(\beta_t - \frac{L\beta_t^2}{2})\|\nabla_{\theta_t} L^c(\hat{w}_t(\theta_t))\|_2^2 \leq L^c(\hat{w}_1(\theta_1)) + \sum_{t=1}^{T}(\alpha_t \rho^2 + \frac{L}{2}\alpha_t^2 \rho^2)$$

$$\sum_{t=1}^{T}(\beta_t - \frac{L\beta_t^2}{2})\min_t \|\nabla_{\theta_t} L^c(\hat{w}_t(\theta_t))\|_2^2 \leq L^c(\hat{w}_1(\theta_1)) + \sum_{t=1}^{T}(\alpha_t \rho^2 + \frac{L}{2}\alpha_t^2 \rho^2).$$

Furthermore,

$$
\begin{aligned}
\min_t \|\nabla_{\theta_t} L^c(\hat{w}_t(\theta_t))\|_2^2 &\leq \frac{L^c(\hat{w}_1(\theta_1)) + \sum_{t=1}^T (\alpha_t \rho^2 + \frac{L}{2}\alpha_t^2 \rho^2)}{\sum_{t=1}^T (\beta_t - \frac{L\beta_t^2}{2})} \\
&\leq \frac{2L^c(\hat{w}_1(\theta_1)) + \sum_{t=1}^T (2\alpha_t \rho^2 + L\alpha_t^2 \rho^2)}{\sum_{t=1}^T (2\beta_t - L\beta_t^2)} \\
&\leq \frac{2L^c(\hat{w}_1(\theta_1)) + \sum_{t=1}^T (2\alpha_t \rho^2 + L\alpha_t^2 \rho^2)}{\sum_{t=1}^T (\beta_t)} \\
&\leq \frac{2L^c(\hat{w}_1(\theta_1)) + \alpha_1 \rho^2 T(2+L)}{T\beta_t} \\
&= \frac{2L^c(\hat{w}_1(\theta_1))}{T} \frac{1}{\beta_t} + \frac{\alpha_1 \rho^2 (2+L)}{\beta_t} \\
&\leq \frac{2L^c(\hat{w}_1(\theta_1))}{T} \max\{L, \frac{\sqrt{T}}{k}\} + \min\{1, \frac{k}{T}\} \max\{L, \frac{\sqrt{T}}{k}\}\rho^2(2+L) \\
&\leq \frac{2L^c(\hat{w}_1(\theta_1))}{c\sqrt{T}} + \frac{k\rho^2(2+L)}{c\sqrt{T}} = \mathcal{O}(\frac{1}{T}).
\end{aligned}
$$

It holds for $\sum_{t=1}^T (\beta_t) \leq \sum_{t=1}^T (2\beta_t - L\beta_t^2)$. In conclusion, it proves that the algorithm can always achieve $\min_{0 \leq t \leq T} \|\nabla_\theta L^c(\hat{w}(\theta_t))\|_2^2 \leq \mathcal{O}(\frac{1}{\sqrt{T}})$ in $T$ steps.

**Lemma 2.** Let $(a_n)_{1 \leq n}, (b_n)_{1 \leq n}$ be two non-negative real sequences such that the series $\sum_{i_i}^\infty a_n$ diverges, the series $\sum_{i_i}^\infty a_n b_n$ converges, and there exists $K > 0$ such that $\|b_{n+1} - b_n\| \leq Ka_n$. Then the seqences $(b_n)_{1 \leq n}$ converges to 0.

*Proof.* See the proof of Lemma A.5 in [Stochastic majorization-minimization algorithms for ].

**Theorem 2.** Suppose the loss function $l$ is L-Lipschitz smooth and have $\rho$-bounded gradients with respect to training data and clean set, and $\lambda(\cdot)$ is differential with a $\delta$-bounded gradient twice differential with its Hessian bounded by $\mathcal{B}$ with respect to $\theta$. Let the learning rate $\alpha_t = \min\{1, \frac{k}{T}\}$, for some $k > 0$, such that $\frac{k}{T} < 1$ and learning rate $\beta_t$ a monotone descent sequence, $\beta_t = \min\{\frac{1}{L}, \frac{c}{\sqrt{T}}\}$ for some $c > 0$, such that $L \leq \frac{c}{\sqrt{T}}$ and $\sum_{t=1}^\infty \beta_t \leq \infty, \sum_{t=1}^\infty \beta_t^2 \leq \infty$. Then

$$
\lim_{t \to \infty} \|\nabla_{w_t} L^{tr}(w_t; \theta_{t+1})\|_2^2 = 0.
$$

*Proof.* It is obvious that $a_t$ satisfy $\sum_{t=0}^\infty a_t = \infty, \sum_{t=0}^\infty a_t \leq \infty$. In Eq. 18, 19, 20, and the linearity of $L$, we rewrite the update of $w$ as

$$
\begin{aligned}
w_{t+1} &= w_t - \alpha_t \nabla L^{tr}(w_t; \theta_{t+1}) \\
&= w_t - \frac{\alpha_t}{n} \sum_{j=1}^n \lambda_j(\theta_{t+1}; w_t) \nabla_{w_t} l(y_j, \hat{y}_j(w_t)) + (1 - \lambda_j(\theta_{t+1}; w_t)) \nabla_{w_t} l(\tilde{y}_j, \hat{y}_j(w_t)).
\end{aligned}
$$

First, we have the difference of the loss function on training set between two iterations,

$$
\begin{aligned}
&L^{tr}(w_{t+1}; \theta_{t+2}) - L^{tr}(w_t; \theta_{t+1}) \\
&= [L^{tr}(w_{t+1}; \theta_{t+2}) - L^{tr}(w_{t+1}; \theta_{t+1})] + [L^{tr}(w_{t+1}; \theta_{t+1}) - L^{tr}(w_t; \theta_{t+1})].
\end{aligned} \tag{33}
$$

For the first term in Eq.33, by the L-Lipschitz-smooth and $\rho-$bounded gradients of $\lambda$ with respect to training and clean set,

$$L^{tr}(w_{t+1}; \theta_{t+2}) - L^{tr}(w_{t+1}; \theta_{t+1})$$
$$= \frac{1}{n} \sum_{j=1}^{n} (\lambda_j(\theta_{t+2}; w_{t+1}) - \lambda_j(\theta_{t+1}; w_{t+1}))l(y_j, \hat{y}(w_{t+1})) + (\lambda_j(\theta_{t+1}; w_{t+1}) - \lambda_j(\theta_{t+2}; w_{t+1}))l(\tilde{y}_j, \hat{y}_j(w_{t+1}))$$
$$\leq \frac{1}{n} \sum_{j=1}^{n} (\left\langle \frac{\partial \lambda_j(\theta; w_{t+1})}{\partial \theta}|_{\theta_{t+1}}, \theta_{t+2} - \theta_{t+1} \right\rangle + \frac{\delta}{2}\|\theta_{t+2} - \theta_{t+1}\|_2^2)(l(y_j, \hat{y}(w_{t+1})) + l(y_j, \hat{y}(w_{t+1})))$$
$$= \frac{1}{n} \sum_{j=1}^{n} (\left\langle \frac{\partial \lambda_j(\theta; w_{t+1})}{\partial \theta}|_{\theta^{t+1}}, -\beta_t \nabla_{\theta_t} L(\hat{w}_t(\theta_t)) \right\rangle + \frac{\delta \beta_t^2}{2}\|\nabla_{\theta_t} L(\hat{w}_t(\theta_t))\|_2^2)(l(y_j, \hat{y}(w_{t+1})) + l(y_j, \hat{y}(w_{t+1}))).$$

For the second term in Eq. 33,

$$L^{tr}(w_{t+1}; \theta_{t+1}) - L^{tr}(w_t; \theta_{t+1})$$
$$\leq \left\langle \nabla_{w_t} L^{tr}(w_t; \theta_{t+1}), w_{t+1} - w_t \right\rangle + \frac{L}{2}\|w_{t+1} - w_t\|_2^2$$
$$= -(\alpha_t - \frac{La_t^2}{2})\|\nabla_{w_t} L^{tr}(w_t; \theta_{t+1})\|_2^2.$$

Therefore, we have

$$L^{tr}(w_{t+1}; \theta_{t+2}) - L^{tr}(w_t; \theta_{t+1})$$
$$\leq \frac{1}{n} \sum_{j=1}^{n} (\left\langle \frac{\partial \lambda_j(\theta; w_{t+1})}{\partial \theta}|_{\theta^{t+1}}, -\beta_t \nabla_{\theta_t} L^c(\hat{w}_t(\theta_t)) \right\rangle$$
$$+ \frac{\delta \beta_t^2}{2}\|\nabla_{\theta_t} L(\hat{w}_t(\theta_t))\|_2^2)(l(y_j, \hat{y}(w_{t+1})) + l(y_j, \hat{y}(w_{t+1})))$$
$$-(\alpha_t - \frac{L\alpha_t^2}{2})\|\nabla_{w_t} L^{tr}(w_t; \theta_{t+1})\|_2^2.$$

Summing up the inequalities in both sides from $t = 1$ to $\infty$, we have

$$\lim_{t \to \infty} \|L^{tr}(w_{t+1}; \theta_{t+2}) - L^{tr}(w_1; \theta_2)\|$$
$$\leq \sum_{t=1}^{\infty} -\frac{\beta_t}{n} \sum_{j=1}^{n} [\|\frac{\partial \lambda_j(\theta; w_{t+1})}{\partial \theta}|_{\theta^{t+1}}\|_2 \|\nabla_{\theta_t} L^c(\hat{w}_t(\theta_t))\|_2 (\|l(y_j, \hat{y}(w_{t+1}))\|_2 + \|l(y_j, \hat{y}(w_{t+1}))\|_2)$$
$$+ \sum_{t=1}^{\infty} \frac{\delta \beta_t^2}{2} \sum_{j=1}^{n} \|\nabla_{\theta_t} L^c(\hat{w}_t(\theta_t))\|_2^2](\|l(y_j, \hat{y}(w_{t+1}))\|_2 + \|l(y_j, \hat{y}(w_{t+1}))\|_2)$$
$$- \sum_{t=1}^{\infty} (\alpha_t - \frac{L\alpha_t^2}{2})\|\nabla_{w_t} L^{tr}(w_t; \theta_{t+1})\|_2^2.$$

Rearrange the terms of the inequality, we obtain

$$\sum_{t=1}^{\infty} \alpha_t \|\nabla_{w_t} L^{tr}(w_t; \theta_{t+1})\|_2^2$$

$$+\sum_{t=1}^{\infty} \frac{\beta_t}{n} \sum_{j=1}^{n} \|\frac{\partial \lambda_j(\theta; w_{t+1})}{\partial \theta}|_{\theta^{t+1}}\|_2 \|\nabla_{\theta_t} L^c(\hat{w}_t(\theta_t))\|_2 (\|l(y_j, \hat{y}(w_{t+1}))\|_2 + \|l(y_j, \hat{y}(w_{t+1}))\|_2)$$

$$\leq \frac{L\alpha_t^2}{2} \|\nabla_{w_t} L^{tr}(w_t; \theta_{t+1})\|_2^2$$

$$+\sum_{t=1}^{\infty} \frac{\delta \beta_t^2}{2} \sum_{j=1}^{n} \|\nabla_{\theta_t} L^c(\hat{w}_t(\theta_t))\|_2^2](\|l(y_j, \hat{y}(w_{t+1}))\|_2 + \|l(y_j, \hat{y}(w_{t+1}))\|_2)$$

$$-\lim_{t\to\infty} \|L^{tr}(w_{t+1}; \theta_{t+2})\|_2 + \|L^{tr}(w_1; \theta_2)\|_2$$

$$\leq \sum_{t=1}^{\infty} \frac{L\alpha_t}{2} \rho^2 + \|L^{tr}(w_1; \theta_2)\|_2 + \sum_{t=1}^{\infty} \frac{\delta \beta_t^2}{2} (2M\rho^2) - \lim_{t\to\infty} \|L^{tr}(w_{t+1}; \theta_{t+2})\|_2$$

$$\leq \infty.$$

The inequality next to last holds since our loss function is bounded by $M$, and the last one holds for $\sum_{t=1}^{\infty} \alpha_t^2$ and $\sum_{t=1}^{\infty} \beta_t^2$ are finite.

In addition, since

$$\sum_{t=1}^{\infty} \frac{\beta_t}{n} \sum_{j=1}^{n} \|\frac{\partial \lambda_j(\theta; w_{t+1})}{\partial \theta}|_{\theta^{t+1}}\|_2 \|\nabla_{\theta_t} L^c(\hat{w}_t(\theta_t))\|_2 (\|l(y_j, \hat{y}(w_{t+1}))\|_2 + \|l(y_j, \hat{y}(w_{t+1}))\|_2)$$

$$\leq 2M\rho\delta \sum_{t=1}^{\infty} \beta_t \leq \infty,$$

we can obtain that

$$\sum_{t=1}^{\infty} \alpha_t \|\nabla_{w_t} L^{tr}(w_t; \theta_{t+1})\|_2^2 \leq \infty. \tag{34}$$

In the other hand, based on the inequality:

$$(\|a\| + \|b\|)(\|a\| - \|b\|) \leq \|a + b\|\|a - b\|,$$

we have

$$\|\|\nabla L^{tr}(w_{t+1}; \theta_{t+2})\|_2^2 - \|\nabla L^{tr}(w_t; \theta_{t+1})\|_2^2\|$$

$$=(\|\nabla L^{tr}(w_{t+1}; \theta_{t+2})\|_2 + \|\nabla L^{tr}(w_t; \theta_{t+1})\|_2)(\|\nabla L^{tr}(w_{t+1}; \theta_{t+2})\|_2 - \|\nabla L^{tr}(w_t; \theta_{t+1})\|_2)$$

$$\leq \|\nabla L^{tr}(w_{t+1}; \theta_{t+2}) + \nabla L^{tr}(w_t; \theta_{t+1})\|_2\|_2 \|\nabla L^{tr}(w_{t+1}; \theta_{t+2}) - \nabla L^{tr}(w_t; \theta_{t+1})\|_2$$

$$\leq (\|\nabla L^{tr}(w_{t+1}; \theta_{t+2})\|_2 + \|\nabla L^{tr}(w_t; \theta_{t+1})\|_2)\|\nabla L^{tr}(w_{t+1}; \theta_{t+2}) - \nabla L^{tr}(w_t; \theta_{t+1})\|_2)$$

$$\leq 2L\rho\|(w_{t+1}, \theta_{t+2}) - (w_t, \theta_{t+1})\|_2$$

$$\leq 2L\rho\alpha_t\beta_t\|(\nabla L^{tr}(w_t, \theta_{t+1}), \nabla L^c(w_t, \theta_{t+1}))\|_2$$

$$\leq 2\sqrt{2}L\rho^2\beta_1\alpha_t$$

$$=C\alpha_t.$$

For Eq. 34 which reads

$$\sum_{t=1}^{\infty} \alpha_t \|\nabla_{w_t} L^{tr}(w_t; \theta_{t+1})\|_2^2 \leq \infty,$$

since $\sum_{t=0}^{\infty} \alpha_t = \infty$, and there exists $K = C > 0$, such that $\|\|\nabla L^{tr}(w_{t+1}; \theta_{t+2})\|_2^2 - \|\nabla L^{tr}(w_t; \theta_{t+1})\|_2^2\| \leq C\alpha_t$, by **Lemma 2.**, we can conclude that

$$\lim_{t\to\infty} \|\nabla_{w_t} L^{tr}(w_t; \theta_{t+1})\|_2^2 = 0,$$

which indicates that the gradient of loss on training set of our algorithm will finally achieve to zero, and thus the iteration of $w$ enables training loss to converge.

