# OpenReview forum: "Towards Robust Graph Neural Networks against Label Noise"
_ICLR.cc/2021/Conference — Reject_

### Official Review · AnonReviewer1 · 2020-10-23
**Interesting work, but can be further improved**

**Rating:** 6
**Confidence:** 4

**Review:**

*quality*
This paper is well-organized. However, the contributions are not clear. Meanwhile, the experimental validation is weak and needs to be improved.

*clarity*
It is not difficult to understand the framework of the proposed algorithm.

*originality*
In this paper, the proposed algorithm focuses on the label noise existing in graph node classification tasks. The contributions are somewhat limited, since this is not the first work focusing on this problem. Besides, the main motivation is also not new for semi-supervised label propagation.

*significance*
Although the proposed algorithm is limited in its novelty, I feel that the significance of this paper will inspire researchers focusing on this domain.

*pros and cons*

Pros:
1.	The topic is practical in realistic machine learning problems, and the proposed method is helpful to reduce the human annotation efforts.
2.	The theoretical study of this paper is meaningful.

Cons:
1.	The authors claim that they are the first to focus on the label noise existing in graph node classification tasks. However, to the best of my knowledge, there already exist some prior works, such as ‘Learning with Inadequate and Incorrect Supervision’,‘Self-Training of Graph Neural Networks using Similarity Reference for Robust Training with Noisy Labels’, ‘Noise-robust semisupervised learning by large-scale sparse coding’, ‘Robust semi-supervised classification for noisy labels based on self-paced learning’, and ‘Label Diagnosis through Self Tuning for Web Image Search’. In this sense, the authors cannot claim that they are the first to solve the label noise in the label propagation or node classification task. Besides, the abovementioned works should be cited, discussed, and even compared.
2.	Some recent works on label propagation can be cited, such as ‘Label propagation via teaching-to-learn and learning-to-teach’, ‘Robust Triple-Matrix-Recovery-Based Auto-Weighted Label Propagation for Classification’, etc.
3.	In Eq. (8), the authors calculate the final predictions by aggregating the original labels and pseudo labels, in order to fully exploit the abundant information contained in the left training nodes D_left. This is a bit confusing, since both the original label y_j and the pseudo label y^\tilde_j may not be the correct labels. The motivation behind this aggregation process should be explained appropriately.
4.	In the experiments, it seems that the author do not use the public splitting for Cora, CiteSeer, and PubMed datasets. As a consequence, the results reported in the paper may not be convincing enough. It would be better to follow the public split for fair comparison.
5.	The proposed model outperforms the comparison methods when the level of flip noise is greater than 0, but cannot perform as good as L2RW when the level equals 0. The reason for such inconsistencies should be analyzed in the experiments.
6.	As for me, the selection of clean labels is a critical step. However, the authors do not give enough details about this process. I suggest analyzing the influence of the scale of clean labels on different datasets.
7.	I cannot find the hyper-parameter configurations for the baselines, and these information should be provided.

---

> ### Author Response · Authors · 2020-11-17
> **#Response to reviewer 4**
>
> Thanks for your insightful reviews and we appreciate your valuable suggestions and recommendations! We address your concerns as follows:
> - Q1: There already exist some prior works focusing on the label noise existing in graph node classification tasks.
> - Answer1: We have cited, discussed and compared with the papers you recommended in the revision, Sec 2.2. We are so sorry that we make an inaccurate description of our contribution. Instead, we are first to address the label noise problem existing in utilizing GNNs to classify graph nodes. This is meaningful because GNNs have been widely applied in various real-world scenarios while high-quality labels are difficult to collect.
> ---
> - Q2: Some recent works on label propagation can be cited.
> - Answer2: We have carefully checked the papers about label propagation you recommended and cited them in the revision, Sec 2.2. Thanks for your recommendation again!
> ---
> - Q3: The motivation behind aggregation of the original labels and pseudo labels should be explained appropriately.
> - Answer3: The reasons why we aggregate the original labels and pseudo labels can be seen as follows. Firstly, as is illustrated in this paper [1], lots of original labels are correct in real world and they designed a compatibility loss to enforce the estimated labels are not completely different from original noisy labels. Secondly, the pseudo labels in our method are soft, not one hot or hard.
> ---
> - Q4: In the experiments, it seems that the author do not use the public splitting for Cora, CiteSeer, and PubMed datasets.
> - Answer4: Firstly, our method is designed to train existing GNNs robustly against label noise, not to propose a new one to compare with existing GNNs so it is unnecessary to correspond to public splitting. Secondly, we have to collect the same nodes per class as clean set, however, clean set with one node per class will be relatively large according to public splitting. And our open-source codes can show our splitting for fair comparison between future methods that train GNNs robustly.
> ---
> - Q5:The proposed model outperforms the comparison methods when the level of flip noise is greater than 0, but cannot perform as good as L2RW when the level equals 0. The reason for such inconsistencies should be analyzed in the experiments.
> - Answer5: Thanks for your reminding! At 0% noise, our method only slightly underperforms L2RW. This is reasonable because the original labels are all correct but our method will inevitably perturb a few labels while the L2RW will not. We have added this explanation in the revision.
> ---
> - Q6: The selection of clean labels is a critical step. However, the authors do not give enough details about this process.
> - Answer6: Only 28,24,24,25 nodes with clean labels in the validation set are provided as the clean set in Cora, Citeseer, Pubmed and Coauthor-Phy datasets and we ensure that each class has the same number of nodes. Besides, we have added some additional experiments for the size of $D_{clean}$ as your advice and we have some interesting findings by this. More details can be found in the revision Sec. 4.5 and Fig. 6. Thanks for your valuable advice again!
> ---
> - Q7: I cannot find the hyper-parameter configurations for the baselines, and these information should be provided.
> - Answer7: We have mentioned hyper-parameter configurations for the baselines in the appendix B, page 14.
> ---
>
> [1] Yi, Kun, and Jianxin Wu. "Probabilistic end-to-end noise correction for learning with noisy labels." CVPR 2019.

---

### Official Review · AnonReviewer4 · 2020-10-27
**Anonymous review**

**Rating:** 5
**Confidence:** 4

**Review:**

This paper presents a label propagation based meta learning algorithm to address label noise. Label propagation helps re-label pseudo labels of noisy data and meta learning achieves aggregations. The method is evaluated on several node classification datasets and a custom version of Clothing1M image classification dataset. The comparison to multiple baselines shows better performance.

Pros:
- The combination of graph neural network and meta learning is interesting and novel

Cons:
- The paper is not well written. Some descriptions are unclear. For instance, the usage of $w$ is never defined. The usage of meta learning is not novel, which from a previous method - L2R, but it is never clearly mentioned. However, it tried to use several equations to explain the L2R method, which is not sufficient to be clear but makes unnecessary confusion.
- Experiments might not be that convincing.
  - The method is only tested on several graph datasets with synthetic noises, which were uncommon in previous papers. However, most compared methods are only evaluated on common image datasets. The comparison may not be very fair. Moreover, from Table 2 and 3. The proposed method is marginally better than several other methods.
  - Although the method tries to tackle a real-world dataset Clothing1M, the scalability issue of this algorithm makes it difficult to work so it is only tested on a custom toy version. So the results are not generally useful to compare to other methods which have tested on full Clothing1M.

---

> ### Author Response · Authors · 2020-11-17
> **#Response to reviewer 3**
>
> We thank reviewer #3 for the helpful and insightful feedback. We provide answers to individual questions below :
> - Q1: the usage of $w$ is never defined.
> - Answer1: $w$ in our paper is the GNNs parameters and we defined it in the revision.
> ---
> - Q2: The usage of meta learning is not novel,it tried to use several equations to explain the L2R method, which is not sufficient to be clear but makes unnecessary confusion.
> - Answer2: Utilizing meta learning to learn the parameters for better learning is common to see. For example, meta-learning with validation set to optimize the discrete graph structure for attack [1]. Combining the formulations with Fig. 2 can help us understand the procedure more clearly. To some extent, this part is similar to re-weight based method. We make a comparison with re-weight based methods (L2R) in the revision Sec 3.3. Instead of learning the weights for samples [2], we utilize meta learning to learn the labels. There are two significant advantages: Firstly, re-weight based methods can not remove the damages caused by incorrect labels because they assign every noisy training sample a positive weight while ours potentially has the ability to take full advantage of noisy samples positively. Secondly, ours can generate comparatively credible labels for other usages while re-weight or some other methods can not.
> ---
> - Q3: Most compared methods are only evaluated on common image datasets. The comparison may not be very fair. The proposed method is marginally better than several other methods.
> - Answer3: The motivation of this work is to utilize the structure information in graph-structured data to train GNNs robustly against label noise. However, there is no intrinsic structure in image dataset. In other words, our method is specially designed for GNNs and graph-structured data. Besides, almost all previous methods are designed for image datasets. Although they can be applied in graph data, they perform not so well in graph data without finetuning. We show their results after removing finetuning in the revision Sec. 4.4 and Fig. 5. The gap between their methods and ours will be enlarged.
> ---
> - Q4: The results are not generally useful to compare to other methods which have tested on full Clothing1M.
> - Answer4: There is no previous method which address the label noise existing in utilizing GNNs to classify graph nodes and there is no graph-structured dataset with real-world label noise like Clothing 1M currently. Besides, GCN is a transductive model which utilize the validation and test nodes during training stage and it is difficult to be applied in large-scale dataset such as Clothing 1M. So we validate the effectiveness of our method to handle real-world label noise on mini-Clothing 1M. Although it is not full dataset, the label noise is still real-world, not synthetic and all the baselines are evaluated on this mini-Clothing 1M for fair comparison. Additionally, we preprocess the Webvision dataset with real-world label noise in the same way with our preprocess on Clothing 1M. The results are reported in the revision, Table 4. Although we validate the real-world effectiveness on mini-Clothing 1M, we believe that our work will motivate future inductive methods that train GNNs robustly against label noise just like the development of GNNs from transductive to inductive models. Also, it is necessary to publish an open-source graph dataset with real-world label noise so that we can evaluate various methods because GNNs have been widely applied in various real-world scenarios while high-quality labels are difficult to collect. Thanks for your valuable advice again!
> ---
>
>
> [1] Zügner, Daniel, and Stephan Günnemann. "Adversarial attacks on graph neural networks via meta learning." ICLR 2019.
>
> [2] Ren, Mengye et al. "Learning to reweight examples for robust deep learning." ICML 2018.

---

### Official Review · AnonReviewer3 · 2020-10-29
**interesting paper but not good enough**

**Rating:** 4
**Confidence:** 5

**Review:**

The paper proposes a robust training algorithm for graph neural networks against label noise. The authors assume the labeled nodes are divided into two parts, clean part without noise and train part with some noise. The proposed method contains two parts. Firstly, it leverages label propagation (LP) trained on the clean nodes to assign pseudo labels on train nodes with noisy labels. Secondly, the authors design a learnable weight \lambda to learn the label for those noisy nodes where LP does not agree with the original labels. The final graph neural network is trained with clean nodes, high confidence train nodes, and uncertain train nodes with learned labels. The authors conduct experiments on four graph datasets with manual injected noise and one real-world noisy dataset to validate the proposed method.

The paper studies an important problem of graph neural networks with noisy label. I have several concerns for the paper.
(1)	The idea of using LP (or another algorithm different from GNN) to create pseudo labels for uncertainty nodes is not a new idea (e.g., in [1]). Actually, the LP, original data and GNN are ensembled and the final label comes from the majority vote. When LP agrees with the original data (in noisy part), those labels are retained.
[1] Li, Qimai, Zhichao Han, and Xiao-Ming Wu. "Deeper insights into graph convolutional networks for semi-supervised learning." arXiv preprint arXiv:1801.07606 (2018).
(2)	Why the joint learning of \theta with GNN parameters is named meta-learning is not very clear. Algorithm 1 shows both \theta and \w_t are fixed to estimate labels for uncertain nodes. Then those parameters are updated with the estimated nodes. Besides, in the experiment part there is no tuned \lambda compared (random is not good, a wrong \lambda can hurt the performance). This makes the improvement from the learnable \lambda less trustful.
(3)	The experiments are less convincing because only one real-world noisy dataset is available. Because the authors have a strong assumption that both noisy and clean labels exist, it would be better if the authors can validate such assumption with some real-world data. The reported numbers show the improvement of proposed method is rather limited, I would suggest run the methods more times and report mean performance with variance. Results with manually tuned \lambda should be reported. Besides, since GNNs are good for learning on graphs with few labeled nodes (such as the gcn paper actually only use several labeled nodes per class), I would suggest adding some additional experiments for the size of D_clean. How will the rest RL + learnable \lambda contribute w.r.t the number of clean nodes?

---

> ### Author Response · Authors · 2020-11-17
> **#Response to reviewer 2**
>
> Thanks for your helpful and insightful feedback! We address your concerns and questions as follows:
> - Q1: The idea of using LP (or another algorithm different from GNN) to create pseudo labels for uncertainty nodes is not a new idea.
> - Answer1: Thanks for your recommendation. We carefully check the paper [1] you recommended and find that it utilized a random walk model to find the most confident vertices and add them to label set to train a GCN. Label propagation in the paper is just a baseline for comparison. Although it is similar to ours in select confident nodes, firstly we utilize label propagation instead of random walk and secondly we select confident nodes in training sets to train GNNs robustly instead of expanding the training sets to explore the global graph structure as paper [1].
> ---
> - Q2: Why the joint learning of \theta with GNN parameters is named meta-learning is not very clear. Algorithm 1 shows both \theta and \w_t are fixed to estimate labels for uncertain nodes. Then those parameters are updated with the estimated nodes.
> - Answer2: We are so sorry that we may describe our method unclearly in our original paper so that you have some misunderstandings of our method. $D_{clean}$ is the initial given a few clean nodes which are not in training set and it is utilized to update $\theta$ in each epoch, not as you claimed to update $\theta$ with the estimated nodes, you can find this in Page 6, Algorithm 1, line 22-23. $\theta$ is the parameters of MLPs which outputs $\lambda$ that is corresponding to a label. We learn the labels that are necessary for better learning, this agrees with the definition of meta learning (“learn to learn” ). We note that you mentioned ‘RL’ in the review. The clean set here serves as the supervision for $\theta$ updating, which is different from reinforcement learning.  If you don’t agree with us, we beg the reviewer to explain the connection between ours and RL.
> ---
> - Q3: In the experiment part there is no tuned \lambda compared. This makes the improvement from the learnable \lambda less trustful.
> - Answer3: We have added two experiments to validate the $\lambda$ optimized by our method are more effective in the revision. Firstly, we compare our $\lambda$ with the tuned $\lambda$ as your advice. We assign the percentage of clean nodes of each label class as $\lambda$. Additionally, we compare the learned $\lambda$ between clean nodes and noisy nodes and find that $\lambda$ of clean nodes are larger on the whole. More details and results can be seen in the revision Sec.4.3, Table 5 and Fig 4.
> ---
> - Q4: The experiments are less convincing because only one real-world noisy dataset is available.
> - Answer4: We have added a dataset (Webvision) with real-world noisy labels in the revision. You can find the results in Table 4.  In fact, it is common to validate the effectiveness of methods handling with real-world label noise on 1or 2 datasets [2,3] because there are not so many open-source datasets with real-world label noise.
> ---
> - Q5: The reported numbers show the improvement of proposed method is rather limited, I would suggest run the methods more times and report mean performance with variance.
> -Answer5: Many baselines that rank the second are after our finetuning on clean set. If we didn’t finetune, their results will be disappointing and the gap between baselines and ours will be enlarged. We have investigated their performance without finetuning in the revision, Sec 4.4 and Fig.5. In fact, we have mentioned that experiment results are repeated 5 times (then average) with different random seeds in the origin paper Sec 4.2. However, we didn’t report the std for the limited space. Now we add it.
> ---
> - Q6: I would suggest adding some additional experiments for the size of D_clean.
> - Answer6: We have added some additional experiments for the size of $D_{clean}$ as your advice and we have some interesting findings by this. More details can be found in revision Sec. 4.5 and Fig. 6. Thanks for your valuable advice again!
> ---
> - Q7: How will the rest RL + learnable \lambda contribute w.r.t the number of clean nodes?
> - Answer7: Our method is meta learning instead of reinforcement learning as our analysis in Answer 2. Besides, we investigate the results w.r.t size of clean nodes in the revision Sec. 4.5 and Fig. 6.
> ---
> [1] Li, Qimai, Zhichao Han and Xiao-Ming Wu. "Deeper insights into graph convolutional networks for semi-supervised learning." AAAI 2018.
>
> [2]Yao, Yu, et al. "Dual t: Reducing estimation error for transition matrix in label-noise learning." NIPS 2020.
>
> [3]Liu, Yun-Peng, et al. "Label Distribution for Learning with Noisy Labels." IJCAI 2020.

---

### Official Review · AnonReviewer2 · 2020-10-29
**The paper is clear and in good quality**

**Rating:** 7
**Confidence:** 4

**Review:**

This paper presents the one technique using label propagation with meta learning. The label smoothness is used to pseudo label the nodes in the graph. The experimental results look promising in two conditions of label noises. The paper is presented clearly and easy to read. Overall, the quality is good.

The paper present the idea and experiments clearly.

I checked a few literature and believe this work is original.

Pros:
1. clear presentation
2. method is simple but seem very effective
3. the experimental results outperformed the state-of-the-arts in both synthetic label noise and real noise scenarios.

Cons:
1. the contributions can be challenged. If check the GNN with label noise, I do can find some literature published on CVPR and ECCV. And the third one is evaluation result, cannot be categorized as a contribution.

---

> ### Author Response · Authors · 2020-11-17
> **#Response to reviewer 1**
>
> Thanks for your insightful reviews and we appreciate your valuable suggestions! We address your concerns and questions as follows:
> - Q1:  If check the GNN with label noise, I do can find some literature published on CVPR and ECCV.
> - Answer1: There are some recent papers combing GNNs with label noise in recent years. But they just utilized GNNs to train DNNs robustly against label noise, not to train GNNs robustly against label noise. For example, the authors of this paper [1] devised a graph convolutional network to correct noisy labels in video anomaly detection; The structure of clean and noisy data is modeled by a graph per class and Graph Convolutional Networks (GCN) are used to predict class relevance of noisy examples in paper [2]. Besides, there is a previous work [3] which presented a noise-tolerant approach for the graph classification task. However, our work is the first to train GNNs robustly against label noise in graph node classification task utilizing the structure information in graph-structured data.
> ---
> - Q2: The third one is evaluation result, cannot be categorized as a contribution.
> - Answer2: We have modified our contribution in the revision and the evaluation results are not categorized as a contribution now. Thanks for your valuable advice again!
> ---
>
> [1]Zhong, Jia-Xing, et al. "Graph convolutional label noise cleaner: Train a plug-and-play action classifier for anomaly detection." CVPR, 2019.
>
> [2]Iscen, Ahmet, et al. "Graph convolutional networks for learning with few clean and many noisy labels." ECCV,2019.
>
> [3] Hoang NT, et al. “Learning Graph Neural Networks with Noisy Labels.” ICLR workshop, 2019.

---

### Author Response · Authors · 2020-11-17
**General response**

We thank all the reviewers for their thorough and very helpful feedback. We have uploaded an updated version based on the comments. The modified parts are shown in red for your convenience. If the title of a subsection is red, the subsection is new. In the meantime, we have provided detailed responses to all your questions below. Please take a look and let us know if you have any further questions or comments!

---

> ### Author Response · Authors · 2020-11-20
> **Summary of revisions made to the paper**
>
>  A summary of the major changes is given below:
> -  We modify our contribution as the first work to handle with the label noise existing in utilizing GNNs to classify graph nodes, which may serve as a beginning for future research towards robust GNNs against label noise.
> ---
> - We describe our meta learning based method more clearly in case that our method is categorized as reinforcement learning.
> ---
> - We add some additional experiments for the size of $D_{clean}$ as the reviewer's advice and have some interesting findings by this. Besides, we add an dataset with real-world label noise called Webvision and two additional experiments to make the improvement from the learnable $\lambda$ more trustful.
> ---
> - We investigate the performance of other baselines  without finetuning when they are applied in graph-structured data and find their performances are disappointing. So we  propose the motivation of this work is to utilize the structure information of graph-structured data to train GNNs robustly against label noise.
> ---
> - We compare our method with re-weight based methods and some robust metods in semi-supervised nodes classification task.
> ---
> We submitted a revised version including the aforementioned revisions. Thank you for all the efforts that help us improve the paper. Looking forward to your valuable reply!

---

### Comment · ~Xuefeng_Du1 · 2021-04-01
**Interesting node classification results on the clean dataset**

Hi,

I notice that the node classification accuracy, when there is no label noise on the Cora dataset, is about 87\%, which is quite interesting! How do you achieve such an accuracy with a similar GCN and training setting to the original ICLR paper? Their result is only 81.5\%.

Thanks,

---

> ### Comment · ~Jun_Xia1 · 2021-04-01
> **Responding to your question.**
>
> Hi Xuefeng,
>
> Thanks for your interest in our work!
>
> As stated in the paper, the split of Cora dataset is 4:4:2，not the same as Kipf's work.  The reason is that we need more training nodes to ensure that clean set is much smaller than noisy training set as previous works about label-noise learning. Besides, all the beselines including vanillaGCN (Kipf's work) are conducted on datasets with this split for fair in this paper.
>
> Any question else?
>
> Best,
>
> Jun.

---

### Decision · Program_Chairs · 2021-01-07
**Final Decision**

**Decision:**

Reject

**Comment:**

We thank the authors for their detailed answers and for providing an updated version of the paper addressing several of the issues raised by the reviewers, including new experimental results.

The paper is technically correct. The comparison with other methods is thorough and includes ablation studies clarifying the contributions of different aspects of the proposed method. One aspect that has been moderately addressed in the new version is the comparison between the "learned lambda" of the paper with a "tuned lambda" suggested by a reviewer. The authors added results where lambda is set to a particular value, however it would be more interesting relevant to consider a real "tuned lambda", i.e., a scalar parameter, shared by all vertices, that is optimized during training; the goal being to clarifying the benefit (if any) of parameterizing lambda as a function of the node, as opposed to a value shared by all nodes.

The paper is clearly written, particularly the revised version.

The novelty is the weakest aspect of the paper. While the specific problem of learning with noisy labels with GCNN may be new, the field of learning with noisy labels in general, and of using label propagation from clean labels to guide the prediction of uncertain labels, has been proposed before, and mentioned in the reviews. The specific instantiation of this idea to the GCNN framework is novel.

The significance of the work is rather positive. The revised version contains results on two real-world datasets, where the proposed method outperforms several existing ones. As mentioned by a reviewer, this paper may inspire other researchers to explore in more depth the specific problems of learning with noise on graphs with GCNN, and to exploit the knowledge of a limited set of clean labels which may have practical importance to reduce human annotation efforts.

In summary, the paper proposes a novel model and demonstrates its potential to address a possibly important problem.
Although the reviewers did not update their reviews, the authors' responses and updated version correctly addresses several of the initial concerns. The limited conceptual novelty compared to existing work did however not convince us to recommend acceptance, given the high selectivity of the conference.

---

> ### Comment · ~Jun_Xia1 · 2021-02-07
> **Response to the decision**
>
> Thank you for your insightful and to-the-point comments and we will further improve this work.